# Sensitivity of the Greenland surface mass and energy balance to uncertainties in key model parameters

Tobias Zolles[1,2] and Andreas Born[1,2]

[1]Department for Earth science, University of Bergen, Bergen, Norway
[2]Bjerknes Centre for Climate Research, Bergen, Norway

**Correspondence:** Tobias Zolles (tobias.zolles@uib.no)

**Abstract.** We investigate the sensitivity of a distributed glacier surface mass and energy balance model using a variance based analysis, for two distinct periods of the last glacial cycle: present day (PD) and the Last Glacial Maximum (LGM). The results can be summarized in three major findings: The sensitivity towards individual model parameters and parameterizations are as invariant in space as in time. The model is most sensitive to uncertainty related to down-welling long-wave radiation. Turbulent latent heat flux has a sizable contribution to the surface mass balance uncertainty in central Greenland today and over the entire ice sheet during the cold climate of the LGM, in spite of its low impact on the overall surface mass balance of the Greenland ice sheet in modern climate. We conclude that quantifying the model sensitivity is very helpful for tuning free model parameters, because it clarifies the relative importance of individual parameters and highlights interactions between them that need to be considered.

*Copyright statement.* TEXT

## 1 Introduction

Of the many challenges to accurately simulate past variations in the volume of the Greenland ice sheet (GrIS) and to project its future contribution to sea level rise, recent studies agree that the uncertainty associated with surface mass balance (SMB) is among the most important (Aschwanden et al., 2019; Plach et al., 2019).

Models to calculate SMB spread a whole range of complexities from empirical index models that only account for air temperature (Ohmura, 2001; Zemp et al., 2019), or temperature and solar radiation (Bintanja et al., 2002; Van Den Berg et al., 2008; Robinson et al., 2011), to coupled atmosphere-snow models that simulate the snow pack in multiple layers and a full representation of the atmospheric circulation, based on physical first principles (Lehning et al., 2002; Fettweis, 2007; Noël et al., 2018). On this spectrum, the empirical models perform well for the observational period and when the temperature sensitivity

of the SMB is well known (Fettweis et al., 2020), but are difficult to constrain for temporal and spatial climate variations and become unreliable for conditions outside their relatively narrow tuning interval (van de Berg et al., 2011; Plach et al., 2019). The lack of constraint in empirical models is unfortunate even though their low computational requirements make them

attractive for the long integration times that are needed to simulate continental ice sheets. Their shortcomings severely limit the usefulness of their results. On the other hand, detailed snow models and especially those coupled with regional atmosphere models are computationally too expensive to run for long periods of time. This situation motivated the development of models that balance the defensible representation of the relevant physical processes with computational efficiency (Krapp et al., 2017; Krebs-Kanzow et al., 2018; Born et al., 2019).

In this study, we use the BErgen Snow SImulator (BESSI), a model that is designed to include all relevant physical mechanisms with reasonable detail but specifically prepared for long integration times by reducing its computational requirements and by strictly conserving mass and energy (Born et al., 2019). Adding to the original model version, we now include three different parameterizations for snow aging based on Oerlemans and Knapp (1998), Aoki et al. (2003), and Bougamont et al. (2005). The turbulent latent heat flux is now simulated also. For this study the model domain was reduced to Greenland at a resolution of 10 km, but all model changes are applicable to the original setup also.

Multiple studies have investigated the impact of the turbulent latent heat flux on the surface mass balance (Box and Steffen, 2001; Box et al., 2004; Van Den Broeke et al., 2008; Cullen et al., 2014; Noël et al., 2018). They find a relatively small impact of the vapor fluxes on the Greenland ice sheet total mass balance ($\approx 5$ Gt a$^{-1}$, Cullen et al. (2014)), but its local impact can be up to 20% of the annual accumulation (Box et al., 2004). The importance of the vapor flux is difficult to assess for different climatic settings, as Box and Steffen (2001) have shown that the choice of the calculation method impacts the results greatly in regions of low flux like the dry interior zone of Greenland. This is exacerbated by the fact that turbulent latent heat fluxes are mostly negative in winter under the present climate conditions, and positive in summer so that the sign of the net flux may change with a different climate. During the colder climate of the glacial a much larger impact of the turbulent latent heat flux can be assumed, similarly to the much greater importance it currently has in Antarctica (e.g., Gallet et al., 2014; Van Wessem et al., 2018). A parameterization based on the bulk-method by Rolstad and Oerlemans (2005) has been added to BESSI to simulate the turbulent latent heat flux.

To assess the sensitivity of the new parameterizations and that of BESSI overall, we employ a variance based approach (Saltelli et al., 2000, 2006, 2010; Sauter and Obleitner, 2015), that has previously been used to quantify the sensitivity of glacier and ice models (Aschwanden et al., 2019; Bulthuis et al., 2019; Zolles et al., 2019). We extend the sensitivity analysis used by Zolles et al. (2019) to provide spatial patterns of sensitivity indices. Following our model's design goal to be used over time scales of glacial cycles and accounting for potentially different sensitivities under different climate boundary conditions, we analyze two large ensembles with a total of 16,500 simulations, for present day (PD) climate and that of the Last Glacial Maximum (LGM). The result is rich information on what parameters and parameterizations have the largest impact on the model's performance, how this sensitivity varies in different regions of Greenland and over time. Knowing the sensitivity also enables a better calibration of the model parameters as the knowledge prevents or reduces over-fitting to a particular study location or time (Beven, 1989).

The revised model is described in section 2. The distributed sensitivity analysis of multiple model output variables to model parameters, including those for the new parameterizations for turbulent latent heat flux and snow aging, is presented in section 3. After that, we discuss our findings in section 4 and conclude in section 5.

## 2 Model description and study setup

The study uses the efficient mass and energy balance model BESSI, which is designed to simulate the mass balance over long time scales (Born et al., 2019). The energy exchange between the snow and the atmosphere is altered in the model version used here, while the subsurface and internal processes are unchanged from the previously published version. The following model was enhanced to include the turbulent latent heat flux and multiple more complex snow albedo schemes were added. The model description given in the following focuses entirely on the interaction between the snow surface and the atmosphere. For the numerical description and other subsurface processes like firnification and heat conduction see Born et al. (2019).

The model setup used here has a 10 km grid for the domain of Greenland. The vertical dimension is discretized based on the mass with up to 15 layers in the snow pack (Born et al., 2019). The mass of each layer is 100 - 500 $\mathrm{kg\,m^{-2}}$. Each cell has a default maximum of 300 $\mathrm{kg\,m^{-2}}$, but due to melt and refreezing the mass may decrease or increase, respectively. Cells above 500 $\mathrm{kg\,m^{-2}}$ or below 100 $\mathrm{kg\,m^{-2}}$ are split or merged respectively, to restore the default maximum value. Simulations require daily input of air temperature, total precipitation, solar radiation and its reference height. Humidity is an optional input which is required if the turbulent latent heat flux is computed. All variables are interpolated to the 10x10 km model grid using bi-linear interpolation. The air temperature is the only meteorological input which is vertically downscaled to the firn model topography using a temperature lapse rate of 6.5 $\mathrm{K/km}$ for PD and 8.55 $\mathrm{K/km}$ for the LGM. The output written by the model may be adjusted by the user ranging from daily over monthly to annual values. Output variables include: surface mass balance, melt of snow, melt of ice, runoff, refreezing, albedo, turbulent latent heat flux, a mask containing snow, land, ice and water as well as the 3-dimensional grid values for snow mass, snow density, snow temperature and liquid water mass.

### 2.1 Surface energy fluxes

The energy exchange between the surface and the atmosphere comprises five different processes, of which the precipitation and the turbulent latent heat flux (vapor flux) also imply a change in mass: The short-wave radiation ($Q_{SW}$), the long-wave/thermal radiation ($Q_{LW}$), the turbulent sensible heat flux ($Q_{SH}$), the turbulent latent heat flux ($Q_L$) and the heat flux associated with precipitation ($Q_P$).

The total surface flux can be expressed as:

$$c_i m_{s,1} \left. \frac{\partial T}{\partial t} \right|_{surface} + Q_M = Q_i + Q_M = Q_{SW} + Q_{LW} + Q_{SH} + Q_L + Q_P \tag{1}$$

where the left hand side denotes the resulting temperature change of the snow/ice ($Q_i$) and the available energy for melting ($Q_M$) if the melting point is reached. Due to the implicit scheme the model uses, no melt is calculated at first, but only energy fluxes and temperatures (even above 273 K). The actual melt is then calculated explicitly each time step as the excess heat above the melting point. The mass flux of the water vapor is also calculated explicitly.

### 2.1.1 Short-wave radiation and albedo parameterization

The energy input to the surface from solar radiation is calculated by using a broadband albedo value:

$$Q_{SW} = (1 - \alpha)SW_{\text{in}}, \tag{2}$$

where $\alpha$ denotes the surface albedo, either $\alpha_s$ for snow or $\alpha_i$ for ice and $SW_{in}$ the incoming short wave radiation at surface height. The albedo value is assumed constant with respect to the solar incidence angle, but undergoes temporal and spatial variations depending on surface properties. We implement four albedo parameterizations of different complexity to simulate the snow albedo. They all have a common maximum albedo value for fresh snow ($\alpha_{fs}$), minimum value for aged snow (firn $\alpha_{fi}$), and ice albedo ($\alpha_i$), but vary in how they calculate the aging.

**1. Constant:** This simple parameterization only uses constant values for dry snow ($\alpha_s = \alpha_{fs}\ T_S < 273K$), wet snow ($\alpha_s = \alpha_{fi}\ T_s = 273K$), and ice ($\alpha_i$). This parameterization has been used before in BESSI (Born et al., 2019).

**2. Oerlemans and Knapp (1998)** This parameterization assumes an exponential decay with time of the fresh snow albedo to a final value of old snow albedo (Oerlemans and Knapp, 1998):

$$\alpha_s = \alpha_{fi} + (\alpha_{fs} - \alpha_{fi})e^{(\frac{t_{fs} - t}{t^*})}, \quad t^* = \begin{cases} 30\text{d}, & T_S < 273.15 \text{ K} \\ 5\text{d}, & T_S = 273.15 \text{ K} \end{cases} \tag{3}$$

where $t_{fs}$ denotes the last day of snowfall, $t$ the current day (time step) with $t^*$ as the characteristic time in days. This or similar parameterizations are usually optimized for the decay rate $t^*$ using observations of albedo or mass balance (e.g., Oerlemans and Knapp, 1998; Klok and Oerlemans, 2004; Bougamont et al., 2005). The very fast decay at the melting point was chosen to account for our very large upper grid box (0.28 - 1.4 m depending on the mass and density of the box), as the heat capacity of the entire large box may delay the melting on the top of the surface layer. Equation 3 does not consider shallow snow packs, where underlying ice or dirty firn albedo may reduce the albedo.

**3. Bougamont et al. (2005)** modified the parameterization by Oerlemans and Knapp (1998) specifically for the Greenland ice sheet by introducing a snow temperature-dependent decay rate:

$$t^* = \begin{cases} 100 \text{ d}, & T_S < 263 \text{ K} \\ 30 \text{ d} + 7\text{d} \cdot (273.15\text{K} - T_S), & 263 \text{ K} \geq T_S < 273.15 \text{ K} \end{cases} \tag{4}$$

Equation 4 results in the same $t^*$ of 30 d as Oerlemans and Knapp (1998) (eq. 3) up to the melting point (273.15 K). This parameterization furthermore introduces an additional wetness-dependent albedo decay in the case of wet snow, which assumes a thin layer of water at the surface according to

$$\alpha_s = \alpha_{fi} - (\alpha_{fi} - \alpha_s) \cdot e^{(\frac{-w_{surf}}{w^*})}. \tag{5}$$

Here $w_{surf}$ denotes the thickness of the water layer and $w^*$ is a characteristic water layer thickness. Since BESSI does not explicitly simulate water at the surface, we adapted the liquid water depending part using a simple linear parameterization. The

decay rate increases depending on the liquid water content $\zeta$ (see 2.2 for details about the liquid water content):

$$t^* = 15 - 14 \cdot \frac{\zeta}{\zeta_{max}} \text{ d}, \; T_S = 273.15 \text{ K} \tag{6}$$

**4. Aoki et al. (2003):** The final albedo parameterization available in the model uses both temperature and time dependent decay rates. There is a linear dependency on temperature in each time step and this parameterization is therefore not exponentially depending on the time since the last snowfall.

$$\alpha_s(t) = \min\{\alpha_s(t-1) - ((T_S - 273.15 \text{ K}) \cdot k + c), \; \alpha_{fi}, \; \alpha_s(t-1)\} \tag{7}$$

where $t$ and $t-1$ are the current and the previous time step, $\alpha_s$ the snow albedo and $k = 1.35 \cdot 10^{-3}\text{K}^{-1}$ and $c = 0.0278$ two empirically based constants. The values are based on averaged values from Aoki et al. (2003) for different spectral bands. To account for a faster decay in a wet snow pack, the albedo is linearly decreased based on the liquid water content of the topmost layer:

$$\alpha_s(t) = \alpha_s(t) - (\alpha_s(t) - \alpha_{fi}) \cdot \frac{\zeta}{\zeta_{max}} \tag{8}$$

If the layer is fully saturated with water the snow albedo instantly drops to its minimum value. This is done in addition to equation 7 at each time step. The albedo increases when new snowfall occurs, but instead of resetting it to the fresh snow value, this albedo is incrementally increased depending on the amount of fresh snow to account for thin layers of snow, and the penetration of short-wave radiation into the older subsurface:

$$\alpha(t) = \alpha(t-1) - (\alpha_{fs} - \alpha_{fi}) \cdot (1 - \exp \frac{-d}{d^*}), \tag{9}$$

where $d$ is the amount of new snow, and the characteristic snow depth $d^*$ is at 3 cm (Oerlemans and Knapp, 1998).

### 2.1.2 Long-wave radiation

The long-wave radiation is a simple parameterization based on the Stefan-Boltzmann law:

$$Q_{LW} = \sigma(\epsilon_{atm}T_{atm}^4 - \epsilon_s T_s^4) \tag{10}$$

where $\sigma$ is the Stefan-Boltzmann constant, $T_{atm}$ and $T_s$ are the 2 m air and snow surface temperature, respectively. The emissivity of snow/ice $\epsilon_s$ is constant at 0.98. Incoming long-wave radiation is only depending on the actual air temperature and the atmospheric emissivity $\epsilon_{atm}$, as the only free model parameter. The lack of confidence in cloud cover of climate models in particular during the last glacial cycle, lead to this decision. Though more complex empirical relations exist (e.g., Listion and Elder, 2006), their applicability for other time scales is questionable. We therefore refrain from using these empirical relationships despite the importance of cloud cover, moisture and aerosols. $\epsilon_{atm}$ varied over a broad range of 0.6-0.9 following the previous configurations of BESSI or similar models (Greuell and Konzelmann, 1994; Busetto et al., 2013; Born et al., 2019). Emissivity values spanning from 0.6 to 0.9 may in reality occur simultaneously in different regions of the Greenland ice sheet, but in the current configuration there is only a single atmospheric emissivity value over the entire ice sheet.

### 2.1.3 Turbulent sensible heat flux

The calculation of the turbulent latent heat flux is based on a bulk method (Braithwaite, 2009) which was applied previously on Greenland as residual method (Rolstad and Oerlemans, 2005). This method assumes a constant turbulent exchange coefficient ($C_h$) for sensible heat over time and space. The only dependency in the previously published parameterization is on the local wind speed $u$ and air temperature $T_{atm}$:

$$Q_{SH} = \rho_{air} c_p C_h u (T_{atm} - T_s) = D_{SH}(T_{atm} - T_s) \tag{11}$$

where $\rho_{air}$ is the density of air and $c_p$ the heat capacity of air. Since BESSI does not use wind speed as an input field, we simplify the equation with a single free model parameter, the turbulent heat exchange coefficient $D_{SH}$ which is subject of the sensitivity analysis. The values given in Table 1 assume an average wind speed of 5 $\mathrm{ms}^{-1}$ if compared to the reported values by Braithwaite (2009). The variation in parameter $D_{SH}$ therefore accounts for both the variability in average wind speed and the efficiency of the exchange $C_h$.

### 2.1.4 Turbulent latent heat flux

The previous version of BESSI did not include turbulent latent heat flux. The new model version includes an optimal turbulent latent heat flux subroutine as part of the setup. The implementation is analog to the turbulent sensible heat flux (eq. 11):

$$Q_L = 0.622 \, \rho_{air} L_v C_h u (e_{air} - e_s) p^{-1} = D_{LH}(e_{air} - e_s) \tag{12}$$

where $D_{LH}$ is the turbulent latent heat exchange coefficient, and $e$ is the water vapor pressure. The parameterization is based on the bulk formulation of Rolstad and Oerlemans (2005) with the latent heat of vaporization $L_v$ and the air pressure $p$. The latter is calculated from the standard pressure at sea level for each grid point. While Rolstad and Oerlemans (2005) assume the same exchange coefficient $C_h$ for vapor and sensible heat, Greuell and Smeets (2001) showed previously that the roughness lengths and the exchange coefficient for momentum and vapor are not necessarily equal. Nevertheless, the parameters $D_{LH}$ and $D_{SH}$ are inherently connected by the surface structure (snow/ice) and the wind speed. To account for the correlation as well as some degree of freedom, our setup uses two free model parameters determining $D_{LH}$, the turbulent exchange coefficient for sensible heat $D_{SH}$ and $r_{lh/sh}$ which are defined by

$$D_{LH} = r_{lh/sh} \cdot 0.622 L_v D_{SH}/c_p \,. \tag{13}$$

In the setup of our study there are three parameters determining the turbulent latent heat flux. The ratio, ($r_{lh/sh}$), accounts for different exchange rates for water vapor and sensible heat. $D_{SH}$ the absolute exchange strength (roughness, wind, stability). The additional parameter, ($\chi_{QL}$), switches the simulation of the turbulent latent heat flux on and off.

### 2.1.5 Precipitation heat flux

The heat supplied by precipitation depends on the atmospheric temperature, which we assume to be in balance with the precipitation. An atmospheric temperature of 273.15 K is the limit of solid precipitation. In the case of snow fall the solid mass of the topmost grid cell of the snow pack increases by the amount of snow and the heat added to this box is

$$Q_{P,\text{snow}} = P\rho_w c_i (T_{atm} - T_s) \tag{14}$$

while rain is added as liquid water mass to the same cell

$$Q_{P,\text{rain}} = P\rho_w c_w (T_{atm} - 273.15\text{K}), \tag{15}$$

where $P$ is the amount of precipitation, and $\rho_w$ and $c_w$ the density and heat capacity of water. If the rain freezes, or percolates further down, the corresponding exchange of latent heat is calculated during the balance calculation as outlined in the next section.

## 2.2 Subsurface percolation and refreezing

Only a brief overview of the subsurface routine is given in this paper. Full details are available in Born et al. (2019). The water holding capacity of each layer determines the percolation. The maximum liquid water content $\zeta_{max}$ is the parameter subject to the sensitivity analysis:

$$\zeta = m_w / m_s \frac{1}{\rho_w (\frac{1}{\rho_s} - \frac{1}{\rho_i})} \tag{16}$$

where $m_w$, $m_s$ and $\rho_w$, $\rho_s$, $\rho_i$ are the masses and densities of water, snow and ice. If the liquid water content $\zeta$ exceeds $\zeta_{max}$ the liquid water mass percolates to the next cell below or is treated as runoff when it leaves the bottom cell.

## 2.3 Global sensitivity analysis

### 2.3.1 Setup and theoretical background

We assess the model sensitivity and uncertainty of BESSI for two different time periods, to verify its applicability over the whole glacial cycle. The PD period using the ERA-interim reanalysis data (Uppala et al., 2011) ranging over 38 years from 1979 to 2017. Our last glacial maximum (LGM, 21,000 years before present) simulations are forced with 30 climate years of the Community Climate System Model version 4 (CCSM4) (Brady et al., 2013). The simulated LGM has an annual mean temperature over the entire model domain of 249 K/-24°C, while the current climate is relatively warm with an annual mean of 262 K/-11°C. The entire model domain was chosen as the ice sheet has different shapes in the two climate states. The precipitation averages are around 400 kg m$^{-2}$ for the LGM and 570 kg m$^{-2}$ for PD. The annual mean solar radiation is about 10% higher for the LGM, though the seasonal cycle deviates drastically from PD. The surface mass balance model uses a static topography, which is based on ETOPO (Amante and Eakins, 2009) for PD and on ICE-6G (Peltier et al., 2015) for the LGM

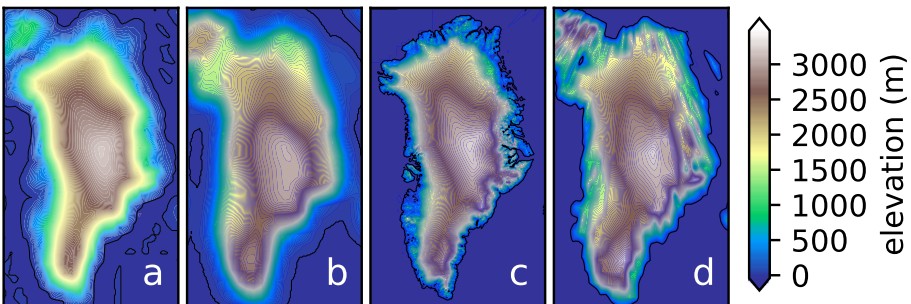

**Figure 1.** The climate model topography for PD (a) is from the ERAinterim data set, and the LGM (b) is from the CCSM4 simulation. The used model topography for PD (c) is ETOPO and ICE-6G (d) for the LGM. For the plot all topography below 5 m was considered sea level for the climate models.

(fig. 1). There is an interpolation artifact in the PD simulation in the far north-west, but as no ice is present in this region of Greenland it does not influence the analysis. BESSI was run for 500 years with the same forcing data looping the forcing data back and forth (1979-2017-1979-2017...) to account for the long response time of the firn cover. After 400 years the firn cover was dynamically (density) and thermodynamically (temperature) stable, even in the regions of very low accumulation. The
analyses shown here are entirely based on the last 100 years of every simulation.

The global sensitivity analysis (GSA) is a variance based method that allows for an assessment of the model sensitivity over the entire parameter space. In contrast to other sensitivity methods, it assesses the full parameter space simultaneously. The method has been applied previously to snow pack (Sauter and Obleitner, 2015) and recently to alpine glacier modeling (Zolles et al., 2019). The method is based on algorithms developed by Saltelli et al. (2000, 2006, 2010), utilizing the setup of
the ensemble hypercube (Sobol et al., 2007). To compute both sensitivity indices, the estimator from Sobol et al. (2007) was used. The probabilistic framework provides an estimate of the sensitivity of the model output to the individual input variables, including parameters and data. The GSA is independent of model calibration and tuning. The model output $Y$, is a function of the input parameters $X_i$: $Y = f(X_1, X_2, .., X_n)$. There are two normalized values that quantify the model sensitivity for each input parameter $X_i$, the first or main order sensitivity index $S_{Xi}$ and the total sensitivity index $S_{Ti}$ of parameter $X_i$. The first
order index denotes the sensitivity of the model towards the parameter $X_i$ only, while the latter includes all the interactions of $X_i$:

$$S_{Xi} = \frac{V_{Xi}(E_{X-i}(Y|X_i))}{V_Y} \tag{17}$$

$$S_{Ti} = \frac{E_{X-i}(V_{Xi}(Y|X_{-i}))}{V_Y}, \tag{18}$$

where $E$ is the expectation value of a given observable such as the SMB. $V_Y$ is the total variance of the given variable and $V_{Xi}$ the variance that only depends on the input parameter $X_i$. $X_{-i}$ denotes the whole parameter space excluding any variation

**Table 1.** The parameter ranges for the free model parameters are rather broad and based on previously published values (Born et al., 2019). The firn albedo may not exceed the fresh snow albedo, and its value was limited to 0.65 during the LGM. All parameters are distributed following a pseudo random Sobol sequence.

| # | Name | Abbreviation | range | unit | reference |
|---|------|-------------|-------|------|-----------|
| 1 | fresh snow albedo | $\alpha_{fs}$ | 0.65 - 0.9 | | Cuffey and Paterson (2010) |
| 2 | firn albedo | $\alpha_{fi}$ | 0.45 - 0.7 (0.65)[1] | | Cuffey and Paterson (2010) |
| 3 | ice albedo | $\alpha_i$ | 0.3 - 0.4 | | Cuffey and Paterson (2010) |
| 4 | turbulent heat exchange coefficient | $D_{SH}$ | 5 - 25 | $\mathrm{Wm^{-2}K^{-1}}$ | Braithwaite (2009) |
| 5 | ratio of sensible and latent heat flux | $r_{lh/sh}$ | 0.5 - 1.2 | | |
| 6 | emissivity of the air | $\epsilon_{atm}$ | 0.6 - 0.9 | | Greuell (1992) |
| 7 | switch for turbulent latent heat flux | $\chi_{QL}$ | on / off | | |
| 8 | albedo module | $\chi_\alpha$ | cnst,Oer,Bou,Aok | | Aoki et al. (2003); Born et al. (2019) |
| | | | | | Bougamont et al. (2005) |
| | | | | | Oerlemans and Knapp (1998) |
| 9 | maximum liquid water content | $\zeta_i$ | 5 - 15 | % pore volume | (Greuell, 1992; Born et al., 2019) |

in $X_i$. The first order index calculates the mean model output ($E_{X-i}(Y|X_i)$) for each representation of $X_i$ and then assesses the sensitivity by calculating the variance for all values of $X_i$.

The total index can be compared to the local sensitivity index that is often determined around the optimal model setting, but the GSA presented here does not rely on a predetermined optimal parameter setting. $V_{Xi}(Y)$ varies the parameter $X_i$ along its dimension, but is computed for all possible points of the parameter space instead of the optimal one. For the detailed algorithm refer to Saltelli et al. (2010). As under-sampling is assumed by the relatively low numbers of simulations, bootstrapping is applied to the ensemble and multiple sensitivity values are reported. Both indices are normalized with the variance of the whole ensemble $V_Y$. We are limiting the detailed discussion to the total index $S_{Ti}$, as BESSI is a highly correlated model.

We are using nine free model parameters (tab. 1). The initial ensemble was generated using a Sobol sequence which consisted of $2000 \times 9$ members for PD and $1000 \times 9$ for the LGM. This sequence spans a 9-dimensional unit hypercube. For computing both sensitivity indices the estimator from Sobol et al. (2007) was used. It splits the initial sequence into two subsets $A$ $B$ each consisting of one half of the initial sequence ($1000/500 \times 9$). Then an additional set of matrices $B_A^i$, which are based on the matrix B where the values for parameter for parameter $X_i$ are replaced with those from subset $A$, are created. The matrices $A$, $B$ and $B_A^i$ are then used to estimate the model sensitivity. A detailed description of the algorithm can be found in (Sobol et al., 2007) and (Saltelli et al., 2010). The whole ensemble consists of $N \cdot (2+k)$ members, with N being the base sample (1000 in the case of PD) and $k$ the amount of parameters (9, tab. 1).

The full ensemble for present day climate has 11000 members, that for LGM climate 5500. The initial hypercube with a length of [0,1] in each dimension is linearly transformed to the intervals given in table 1, with the exception of the latent heat flux switch and the albedo module. These two parameters have two, respectively four discrete values, and the parameter

space is split equally between them. The model simulations are carried out with the generated parameter matrix. We are using bootstrapping to estimate the sensitivity indices. For each bootstrap the equations 17 and 18 are evaluated. Finally, we report the mean sensitivity indices and their standard deviation. The results were checked for consistency ($\sum S_{Xi} \leq 1$, $S_{Xi} \leq S_{Ti}$). The ensemble size used during the LGM is on the absolute lower limit of applicability for GSA, as the standard deviation of the sensitivity indices is large (fig. A1). The GSA works well for the SMB, latent heat flux and melt, some with increased uncertainty, but fails for the 10 m firn temperature for example. Due to the larger ensemble the confidence in PD ensemble is higher, but not by a large enough margin to justify the additional computation time relative to the 5500 members of the LGM ensemble. BESSI is a complex model with all parameters interacting and $S_{Xi}$ provides less information. Therefore, the results mainly focus on the total sensitivity index $S_{Ti}$.

The GSA was computed for five different outputs: albedo, vapor flux, snow melt, surface mass balance and surface temperature, which are based on average yearly sums for SMB, melt and vapor flux or temporal averages for albedo and temperature over 100 years. Surface temperature results are rather uncertain and only tendencies can be extracted as the surface temperature is largely influenced by the annual cycle and fresh snow fall on an ice surface.

## 3 Results

### 3.1 Global sensitivity analysis - GSA

**GSA at PD over elevation bands:** The main focus of the results is on the surface mass balance and discussion is limited to the total sensitivity index ($S_{Ti}$), due to limited information that can be extracted from $S_{Xi}$ in complex models. The sensitivity of the SMB for different elevation classes for the present day ice sheet is shown in Figure 2. In the region from 0 - 1000 m the largest sensitivity is associated with the parameter uncertainty of the atmospheric emissivity $\epsilon_{atm}$ with a normalized total sensitivity index of about 0.8. The second most influential parameter is the turbulent heat exchange coefficient $D_{SH}$, followed by a slight influence of the snow albedo (fig. 2 a). The general features are similar up to 2000 m (fig. 2 b), with a slight decrease in the sensitivity to $D_{SH}$. In regions above 2000 m (fig. 2 c,d,e) the SMB is sensitive to a much wider range of parameters: the atmospheric emissivity, the fresh snow $\alpha_{fs}$ and firn albedo $\alpha_{fi}$, the choice of albedo module $\chi_\alpha$, the turbulent heat flux coefficient, and the turbulent latent heat flux switch $\chi_{QL}$. While the importance of $\chi_{QL}$ is increasing with elevation, the importance of $\alpha_{fi}$ decreases. The sensitivity of the fresh snow albedo $\alpha_{fs}$ increases up to 0.4 at 2500, but not further. With the increasing sensitivity of multiple parameters the relative sensitivity of $\epsilon_{atm}$ decreases, leading to $\chi_{QL}$ almost being equally important in the region with the highest average elevation. The local SMB in regions above 2000 m is impacted by multiple parameters, while the integrated mass balance is dominated by the atmospheric emissivity with the snow albedo and turbulent fluxes having a minor influence.

**Spatial pattern of GSA at PD:** The global sensitivity maps for all parameters are displayed in Figure 3. The GSA was calculated for each grid cell individually. The general trends are similar to the elevation averages, but the spatial pattern shows important additional details (fig. 2). The SMB is sensitive to $\epsilon_{atm}$ over the entire ice sheet. The SMB is also sensitive to $\chi_{QL}$ in the interior of Greenland, and $\chi_{QL}$ can be considered a sensitive parameter for most of the ice sheet apart from the regions of

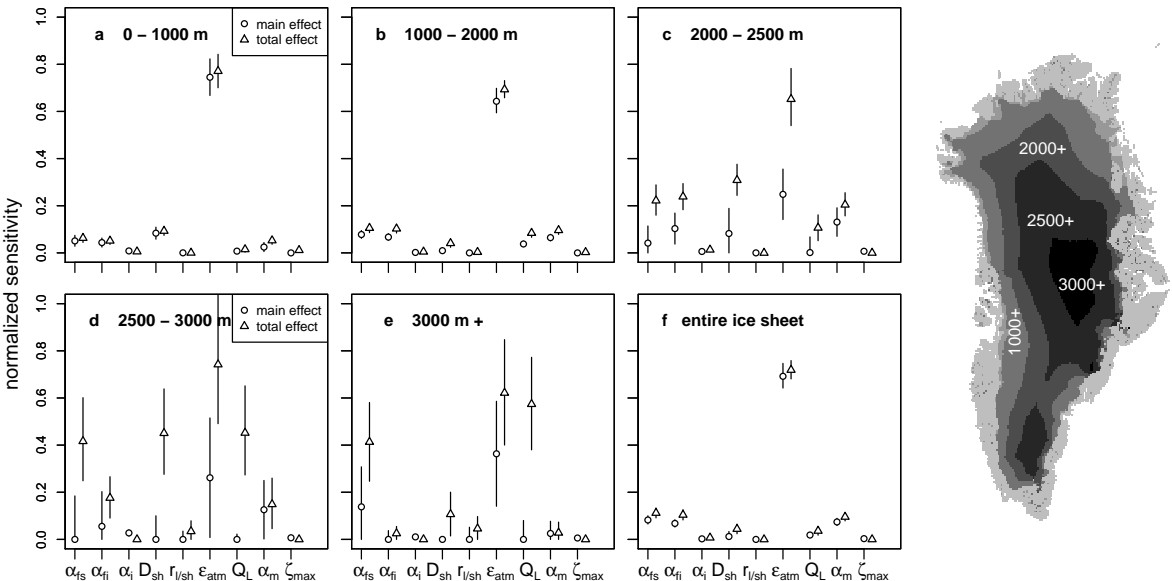

**Figure 2.** The global sensitivity analysis of the PD-SMB provides the main order effect (circle) and the total effect (triangle). The two indices are displayed for all nine parameters over the different elevation bands ranging from 0-1000, 1000-2000, 2000-2500, 2500-3000, above 3000 m and the entire ice sheet. The symbol represents the mean value of the sensitivity index with the bars as $\pm\,1\sigma$. The elevation bands of the present day topography over Greenland are displayed on the right, the analysis is only done for cells where ice is present. A similar figure for the LGM is found in the appendix (A. A1).

very high melt. In the interior of very high elevation, the most sensitive of the three snow albedo related parameters is the one for fresh snow $\alpha_{fs}$, while at elevations below 2000 m the firn snow albedo $\alpha_{fi}$ and the chosen type of albedo parameterization, $\chi_\alpha$, is more important, the latter in particular in the north-east, where fresh snow fall is infrequent. The SMB is sensitive to $D_{SH}$ at the ice caps in the west and on the ice sheet above 1500 m, only at the top of the ice sheet its influence is reduced. The
ice albedo $\alpha_i$ plays a minor role in the north, but is generally of very low impact. Additionally, $\zeta_{max}$ as well as $r_{lh/sh}$ are of minor importance.

    The dominance of $\epsilon_{atm}$ is a result of the change in total heat flux associated with its parameter uncertainty. At an annual average air temperature of -10°C, a change of the atmospheric emissivity from 0.6 to 0.9 increases the heat flux by 80 $\mathrm{Wm}^{-2}$, while a change of albedo from 0.9 to 0.6 does only increase the energy input by 40 $\mathrm{Wm}^{-2}$ for typical values of solar radiation
annual averages. Similarly, the sensible heat flux is smaller than the other heat fluxes over most of the ice sheet (monthly averages from -20 to +50 $\mathrm{Wm}^{-2}$ with $D_{SH} = 12\,\mathrm{Wm}^{-2}\mathrm{K}^{-1}$) so even a doubling will not be larger than the change of atmospheric emissivity on an annual basis. The relatively low impact of the ice albedo $\alpha_i$ is due to the small exposure time and the small parameter range. An ice albedo change of 0.3 to 0.4 will at most result in an annual average energy change of 10-20 $\mathrm{Wm}^{-2}$, but the ice is never exposed for the whole season. Rather the date of ice exposure, which is a result of the energy fluxes prior to its
exposure and associated with an surface albedo change from 0.55-0.9 to 0.3-0.4, is much more important. The SMB in regions

# Sensitivity of the SMB in PD

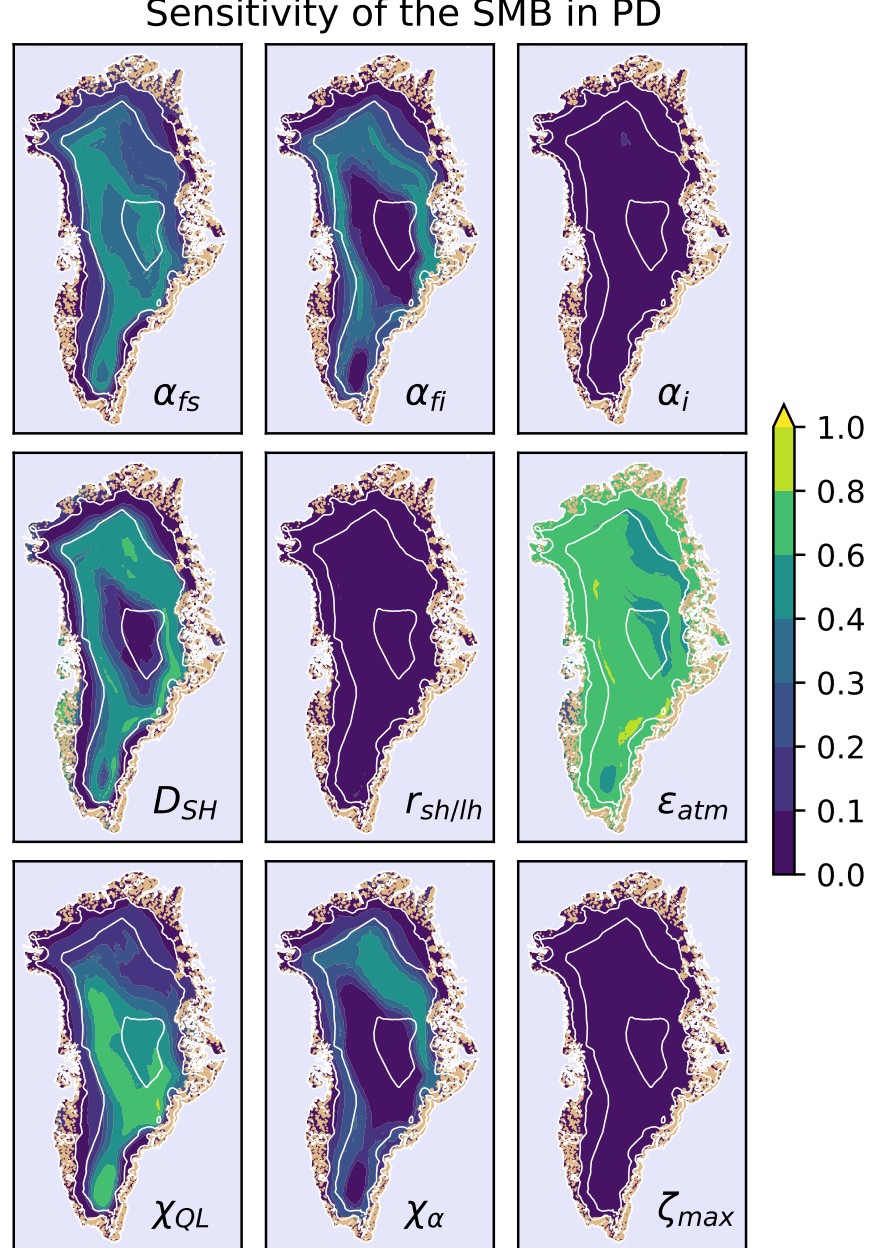

**Figure 3.** Global sensitivity at PD. The total sensitivity index of the SMB of every parameter for PD is displayed for every ice covered grid cell. The ice free land is in brown, the ocean in light blue, 1000 m contours are in white. The sensitivity is largest for the atmospheric emissivity $\epsilon_{atm}$, followed by the fresh snow and firn albedo $\alpha_{fs/fi}$, the turbulent heat exchange coefficient $D_{SH}$, and the latent heat flux switch $\chi_{QL}$. The SMB is not sensitive to the ice albedo $\alpha_i$, the ratio of the turbulent exchange coefficients $r_{lh/sh}$ and the liquid water content $\zeta_i$. The maps do not include uncertainty, but the uncertainties of the sensitivities are in the same order of magnitude as shown in Figure 2.

above 1000 m is sensitive to the snow albedo, which in turn is a function of snowfall, snow temperature and the chosen albedo parameterization. Each albedo model treats these processes differently. While the basic one only distinguishes between dry and wet snow, the others account for snow aging ranging from time over time-and-temperature to time-temperature-wetness dependency. The choice of albedo model is not important in the interior of Greenland, as temperatures are low, albedo decay is slow and the snow does not get wet. At slightly lower elevations there is an interplay of the fresh snow $\alpha_{fs}$ and firn albedo $\alpha_{fi}$ and the chosen decay parameterization $\chi_\alpha$, with a larger impact of the fresh snow albedo, as snowfall is not frequent and decay rates at around -10 C are of the order of a few weeks (Sec. 2.1.1). An exception is found in the north east, which is characterized by the driest climate on Greenland. The less frequent precipitation, and therefore albedo resetting explains a larger dependency on the decay rate and the choice albedo module. The north-east of Greenland is furthermore an area higher up on the Greenland ice sheet where $\chi_{QL}$ is less important. Due to the low amount of precipitation the SMB is quite sensitive to changes in the surface energy balance (SEB). Ensemble members with very high energy balance (low albedo, high emissivity) lead to a melting state, which result in very large changes in SMB relative to the small changes associated with condensation and sublimation. The large impact of $D_{SH}$ at the western coast is due to the large air-surface temperature difference, while in the interior Greenland the atmospheric temperature is much closer to the snow surface temperature.

**Spatial pattern of GSA at LGM:** The sensitivity of the SMB at the LGM shows similar features as at PD, but shifted to lower elevations (fig. 4). During the much colder and dryer LGM the lowest region shows an increased importance of the choice of the snow albedo module. The turbulent latent heat flux switch $\chi_{QL}$ is as important as the atmospheric emissivity $\epsilon_{atm}$ already above 1000 m and the fresh snow albedo $\alpha_{fs}$ is almost as important above 2000 m. The ice sheet integrated SMB shows strong sensitivity to atmospheric emissivity ($S_{Te_{atm}} \approx 0.3 - 0.7$), the turbulent heat flux coefficient (up to 0.8 coastal), the latent heat flux switch ($S_{T_{\chi_L}}$ mostly $> 0.4$) and the snow albedo ($S_{T_\alpha} \approx 0.3$) (fig. 4) during the LGM. The liquid water content $\zeta_{max}$, the ice albedo $\alpha_i$ and the ratio of latent and sensible heat exchange coefficient $r_{lh/sh}$ do only marginally impact the SMB in either of two climate states (fig. 3 and 4). On the local scale $\chi_{QL}$ followed by $\epsilon_{atm}$ and $\alpha_{fs}$ are the main sensitivity components at the LGM (fig. 4). The sensible heat flux exchange coefficient $D_{SH}$ is important along the margin with the largest impact in the south-east. The firn albedo and the albedo module are sensitive parameters along the margin, apart from the precipitation heavy south and southeast, where frequent snowfall resets the albedo.

The increased importance of $\chi_{QL}$ is a result of a much colder and dryer climate. The SMB is positive over most of Greenland during the LGM, even for parameter combinations leading to a high energy input. In the absence of melt, the only change to the SMB is due to sublimation and hoar formation. The dry climate also favors higher sublimation than at PD. The vapor flux (sublimation) is mainly a net heat loss for the surface at the LGM. The surface temperature via the Clausis-Clapeyron relation has an exponential impact on the turbulent latent heat flux resulting in a greater model sensitivity towards the atmospheric emissivity than the actual exchange coefficient $D_{SH}$ (eq. 13). The incoming long-wave radiation is also the largest energy source for the surface. The large impact of $D_{SH}$ in the south-east is due to the large temperature difference between the surface and the air. The south-east is dominated by intense precipitation and rather warm air masses even during the LGM. There is a precipitation gradient from the western coast to central Greenland, the least precipitation is found in the west of Greenland and the south of Elsmere Island due to the presence of the Laurentine ice sheet over North America at the LGM (not shown).

# Sensitivity of the SMB in LGM

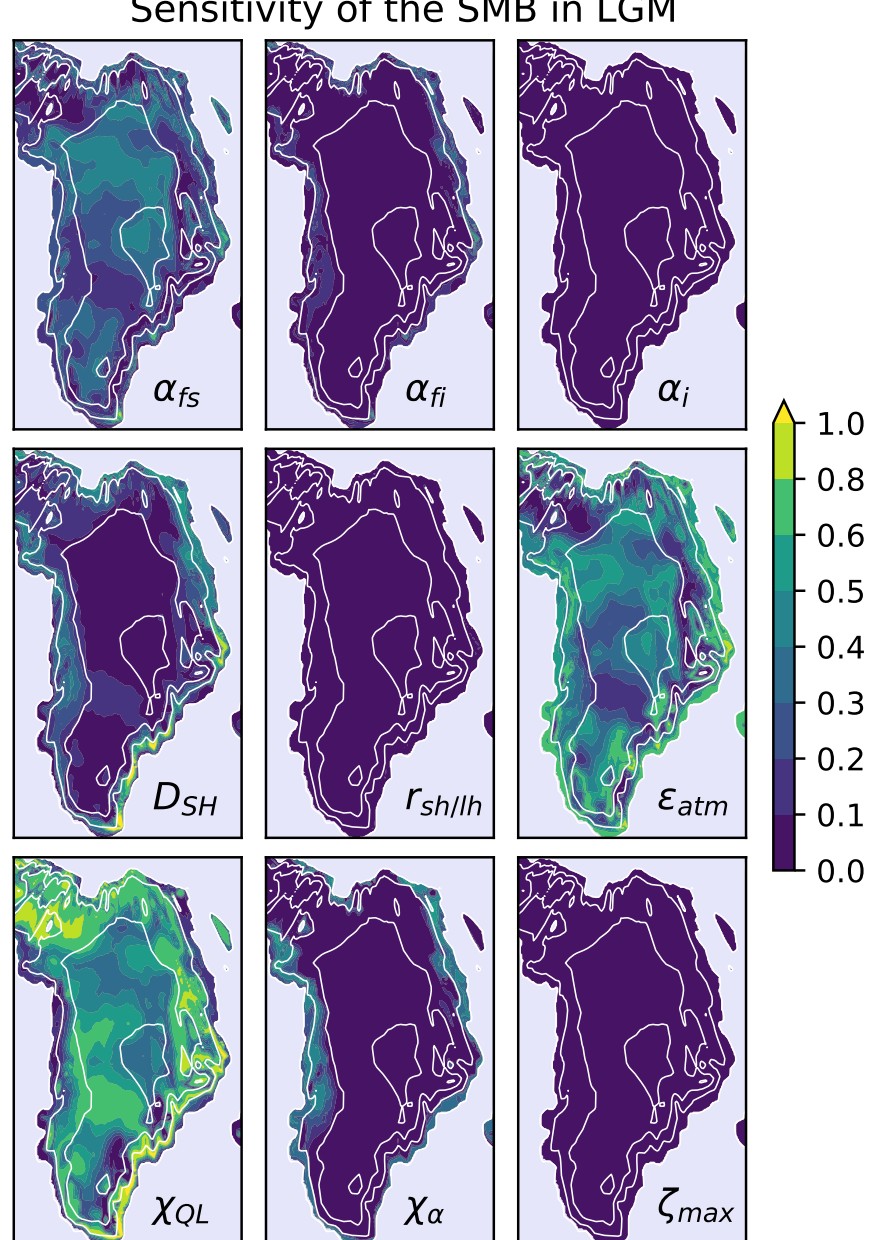

**Figure 4.** Global Sensitivity at the LGM. The total sensitivity index of the SMB of every parameter for PD is displayed for every ice covered grid cell. The ice free land is in brown, the ocean in light blue, 1000 m contours are in white. The total sensitivity index of the surface mass balance for the LGM is more variable than during PD. The model shows the greatest sensitivity towards $\chi_Q$ and the atmospheric emissivity $\epsilon_{atm}$ followed by the fresh snow albedo $\alpha_{fs}$. Firn albedo $\alpha_{fi}$ and the albedo module $\chi_\alpha$ as well as the turbulent exchange coefficient $D_{SH}$ are important around the margin. The SMB during the LGM shows almost no sensitivity towards the other parameters.

Therefore, the albedo module is more important on the western than the eastern margin. As frequent precipitation, which is mainly snowfall during the LGM, increases the albedo more frequently. The lower model sensitivity at the LGM towards $\epsilon_{atm}$ is mainly a result of the lower air temperature with annual averages being around 10 K lower, resulting in less incoming long-wave radiation and a lower absolute impact of the emissivity.

**Sensitivity of other output variables in addition to SMB:** We also studied the sensitivity of other model variables, namely the surface albedo, turbulent latent heat flux, snow melt, and surface temperature. Due to their large size, the corresponding figures (similar to fig. 3 & 4) are included in the supplement, but we include the main findings here. The global sensitivity for the annual average albedo during PD period is mainly influenced by the fresh snow albedo parameter, with only minor importance of the firn albedo, ice albedo, the choice of the albedo module, and the atmospheric emissivity at the ice sheet

margin. As a result, the snow albedo should not be tuned with BESSI without incorporating the atmospheric emissivity in addition to the direct albedo related parameters (fig. S. GSA_albedo_ERAi/LGM).

Besides the switch ($\chi_{QL}$) which disables the turbulent latent heat flux $Q_L$ completely, the turbulent latent heat flux is most sensitive to the atmospheric emissivity $\epsilon_{atm}$ followed by the turbulent heat flux exchange coefficient $D_{SH}$, mainly around the margins (fig. A A3). Around the margin the ratio of turbulent sensible and latent exchange coefficients $r_{lh/sh}$ plays a minor

role ($S_{ti} \approx 0.1$). The turbulent latent heat flux is also sensitive to the snow albedo related parameters ($\alpha_{fs/fi}, \chi_\alpha$) in the north. The turbulent latent heat flux $Q_L$ is neither sensitive the maximum liquid water content $\zeta_{max}$ globally nor to the ice albedo and $r_{lh/sh}$ in the interior of the ice sheet. As the effect of $r_{lh/sh}$ on the turbulent latent heat flux as well as the SMB is low, using similar exchange coefficients for moment, temperature and water vapor are justified within this framework. At the LGM, $Q_L$ shows an increased sensitivity to $\alpha_{fs}$, while the $D_{SH}$ is less important around the margin, due to lower atmospheric

temperatures and slower albedo decay (not shown). The albedo module and the firn albedo play almost no role in either case.

The average snow temperature is mainly influenced by $\epsilon_{atm}, \alpha_{fs}, D_{SH}, \chi_{QL}$, though uncertainties of the sensitivity are rather large, due to temperature resetting in the event of snowfall on ice or shallow snow packs.

Snow melt is closely linked to the SMB, and shows similar sensitivities as reported for the SMB. Just as expected, the impact of the latent heat flux switch is less as it is mainly important in regions with the absence of melt (fig. 3).

## 3.2   Ensemble statistics

**PD regional parameter dependencies:** The GSA clearly highlights the surface mass balance as most sensitive to the atmospheric emissivity and the latent heat flux switch irrespective of background climate. The GSA method has the drawback that it gives no information about the sign and absolute magnitude of the SMB changes. Physical processes and associated parameters, which result in either surface heating or cooling, will be analysed based on ensemble statistics explained in the following.

Conversely, an increase in albedo and a decrease in atmospheric emissivity always lead to an increase in surface mass balance. The ensemble statistics also gives additional information for parameter sensitivities which were well analyzed with GSA. We split Greenland into 11 different regions for PD and 13 for the LGM (2 more around Elsmere Island) based on on elevation, ice divides, geography and climatological similarity for this analysis. In particular most of the west coast shows a similar behavior and is therefore only a single region (fig. 5). This is a regional spread of the elevation bands used in Figure 2. For each

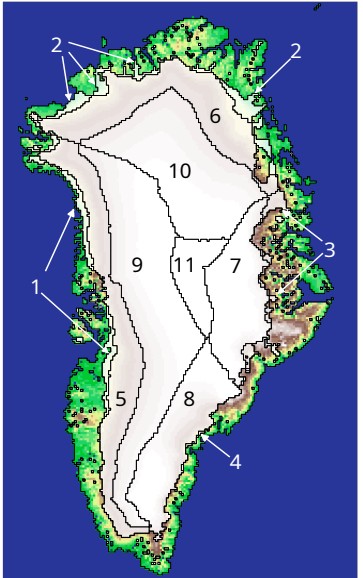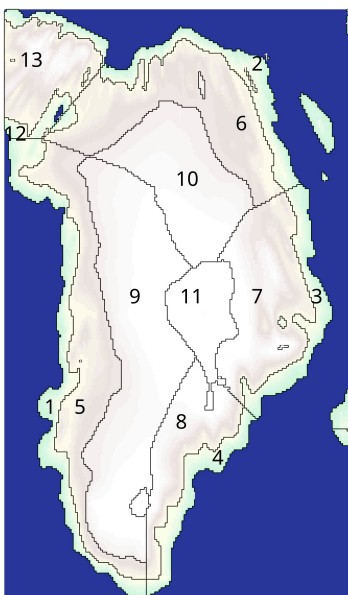

**Figure 5.** Greenland is split into 11 regions based on elevation, geographic and climatic similarity for PD (left) and the LGM (right). There are four different sections with SW-W, N, E, SE and the area around Ellesmere island for the LGM. Each section is split into three elevation bands ranging from 0-1000-2000-3000 m. During the present day three regions in the west and north from 0-1000-2000-3000 m (1/5/9, 2/6/10). The southeast and east regions are precipitation driven and the change in SMB with altitude is less developed, therefore 1000-3000 m is joined to one region (4/8, 3/7). There is one additional region in the center which is at elevations above 3000 m (11). Ice covered areas are in white with a slight elevation shading in the background. The ice free area is based on elevation coloring with green as the lowest and brownish the highest.

region the parameter range is split into 20 equally spaced intervals for which the 5, 25, 33, 50, 66, 75, and 95% quantiles are calculated. The binning is necessary as the ensemble was not created with parameters at regularly spaced intervals. The most interesting features are seen for the parameters where the effect changes sign depending on the atmospheric conditions, like the turbulent heat exchange coefficient. We discuss the selected region 5 and the turbulent latent heat flux in depth here, while
5   the complete set of figures is included in the supplement for reference.

    In Figure 6 the impact of the various parameters on the PD SMB is shown for region 5, the western region of Greenland ranging from 1000 - 2000 m. The dominant parameter is the atmospheric emissivity $\epsilon_{atm}$. Over the range of plausible values $\epsilon_{atm}$ reduces the median of the SMB by almost 800 $\mathrm{kgm}^{-2}$. The atmospheric emissivity and the SMB are inversely correlated and the relationship is non-linear with greater effect at larger values. The spread of the ensemble, i.e., the variance of the SMB
10   as a result of other parameters, increases too. The increase of the SMB with the snow albedo related parameters $\alpha_{fi}$ and $\alpha_{fs}$ is smaller and the width of the distribution decreases (panel a,b), as even very low albedo parameter values do not necessarily lead to a negative SMB in the western region. An increase in $D_{SH}$ slightly reduces the median SMB, but the spread is decreasing. Ice albedo $\alpha_i$, liquid water content $\zeta_{max}$ and $r_{lh/sh}$ have almost no impact. The SMB increases in the presence of turbulent

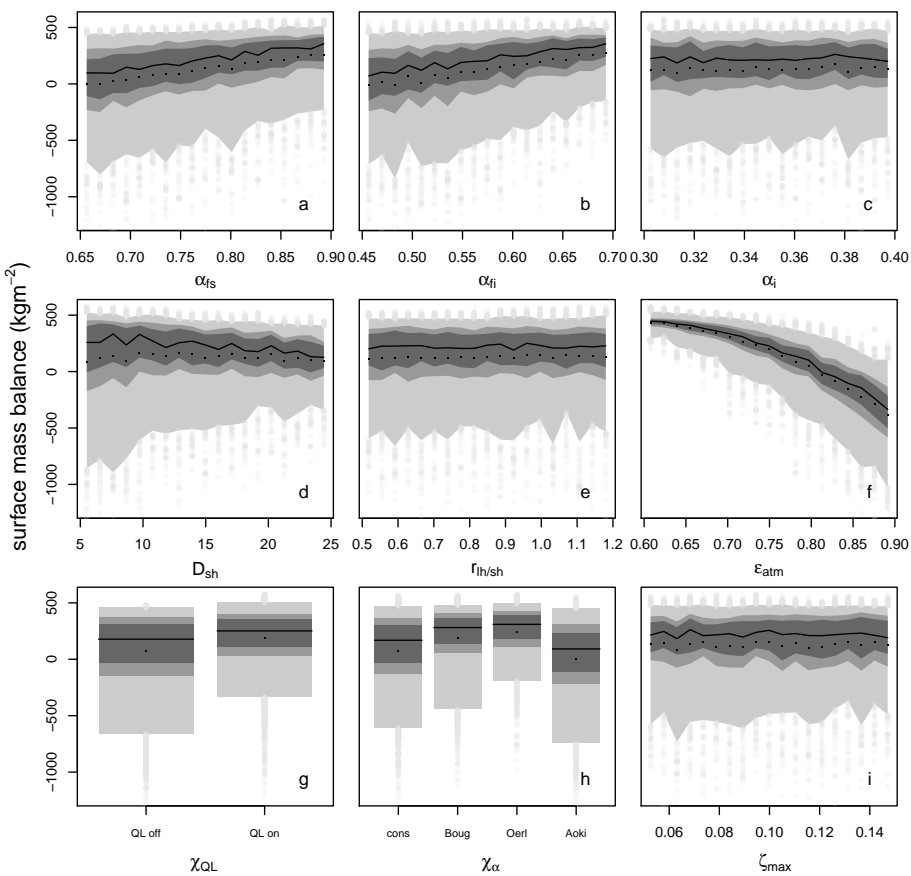

**Figure 6.** The ensemble statistics for the surface mass balance for region 5 (west 1000-2000 m) at PD. The 5/95, 25/75, 33/66 and 50 quantiles are displayed in progressively darker shading. Black points represent the ensemble mean and the grey points correspond to the rest of the ensemble, apart from outliers (max 5 per bin allowed), which are removed to improve readability. Each plot represents the range of one parameter with $\alpha_{fs}$, $\alpha_{fi}$ and $\alpha_i$ in the top row, $D_{SH}$, $r_{lh/sh}$ and $\epsilon_{atm}$ in the middle and $Q_{Lon/off}$, $\chi_\alpha$ and $\zeta_{max}$ at the bottom. As $Q_{Lon/off}$ and $\chi_\alpha$ have two and four discrete values, the parameter range is not split in 20 intervals.

latent heat flux due to the heat loss of sublimation in region 5. With all albedo modules the ensemble has a wide spread, but the variation is smallest for the time dependent decay (Oerlemans and Knapp, 1998) which also has the highest median mass balance, as it neither has an instant albedo drop upon reaching the melting point (constant) nor accounts for the liquid water mass in the snow pack. The parameterization based on temperature, wetness and time has the lowest median SMB.

The strong impact of the emissivity on the surface energy balance and therefore also SMB is due to the larger annual average of the incoming long-wave radiation relative to the short-wave, precipitation, and turbulent fluxes. In the PD climate, the largest energy source for the snow is incoming long wave radiation. As atmospheric emissivity $\epsilon_{atm}$ decreases less energy is available for melt in region 5. In the absence of melt, surface mass balance response is only due to the sublimation and since the vapor flux has a modest absolute impact on the SMB, the spread of the ensemble is low. Vice versa, at large $\epsilon_{atm}$

values warming, and potentially early melting of the snow pack, occurs in many grid cells leading to a positive feedback effect with lower albedo. In agreement with the GSA (fig. 3) the firn albedo $\alpha_{fi}$ is almost as important as fresh snow albedo $\alpha_{fs}$. The ELA is located in region 5 and snow temperatures are rather warm, resulting in a large impact of snow albedo decay and its parameterization. The constant albedo parameterization spreads more than $1000\,\mathrm{kg\,m^{-2}}$, as the albedo is very sensitive around the ELA, changing instantly from $\alpha_{fs}$ to $\alpha_{fi}$ when the snow-pack reaches the melting point. While the albedo

parameterizations based on Oerlemans and Knapp (1998) and Bougamont et al. (2005) have a temporal decay relative to the instant one in the constant case. Bougamont et al. (2005) has a snow temperature dependent increased decay rate and even more in the wet case, leading to less skewed ensemble than of the other albedo parameterizations. The last albedo parameterization decays faster than the other two at warmer temperatures, leading to lower albedo than all other albedo parameterizations before the melting point is reached. Additionally, there is a difference between the models in case of snowfall.

**Differences of the LGM regional parameter dependencies:** During the LGM the western region between $1000 - 2000\,\mathrm{m}$ (not corresponding to the identical geographical area as PD due to topographic differences) shows positive, but low SMB with a lower spread of the ensemble (fig. 7, A A2). There are three distinct differences to PD: The impact of $\alpha_{fi}$ and $\chi_\alpha$ is drastically reduced, enabling the latent heat flux results in a decrease of SMB, as sublimation prevails at the LGM and reduces the SMB, and the importance of $\alpha_{fs}$ increases relative to $\epsilon_{atm}$ (fig. 7). The climate in the west of Greenland in the LGM is

characterized by a much lower air temperature, slightly more annual mean radiation and lower precipitation. The lower air temperature reduces the impact of $\epsilon_{atm}$ and produces a more positive SEB, which in turn results in lower snow temperatures leading to a slower snow albedo decay (for the albedo subroutines which parameterize the decay), no melting and almost no impact of associated parameters. The incremental increase of snow albedo with snowfall, gives slightly lower albedo in the dry climate. During the LGM sublimation prevails over the entire year. Though in the absence of melt the increase in sublimation

results on a mass loss, rather than the reduced melt via cooling due to sublimation during the PD conditions. Furthermore, the non-linearity in the SMB at the LGM almost vanishes for $\epsilon_{atm}$ and $\alpha_{fs}$ as there is hardly any melt and associated snow-ice albedo feedback.

**Impact of the turbulent latent heat flux exchange coefficient on the SMB as an example for PD:** The turbulent sensible heat flux may either be a heat loss or heat gain for the surface depending on the difference between atmospheric and surface temperature.

temperature. The strength of the turbulent sensible heat flux is mainly influenced by the turbulent latent heat flux exchange

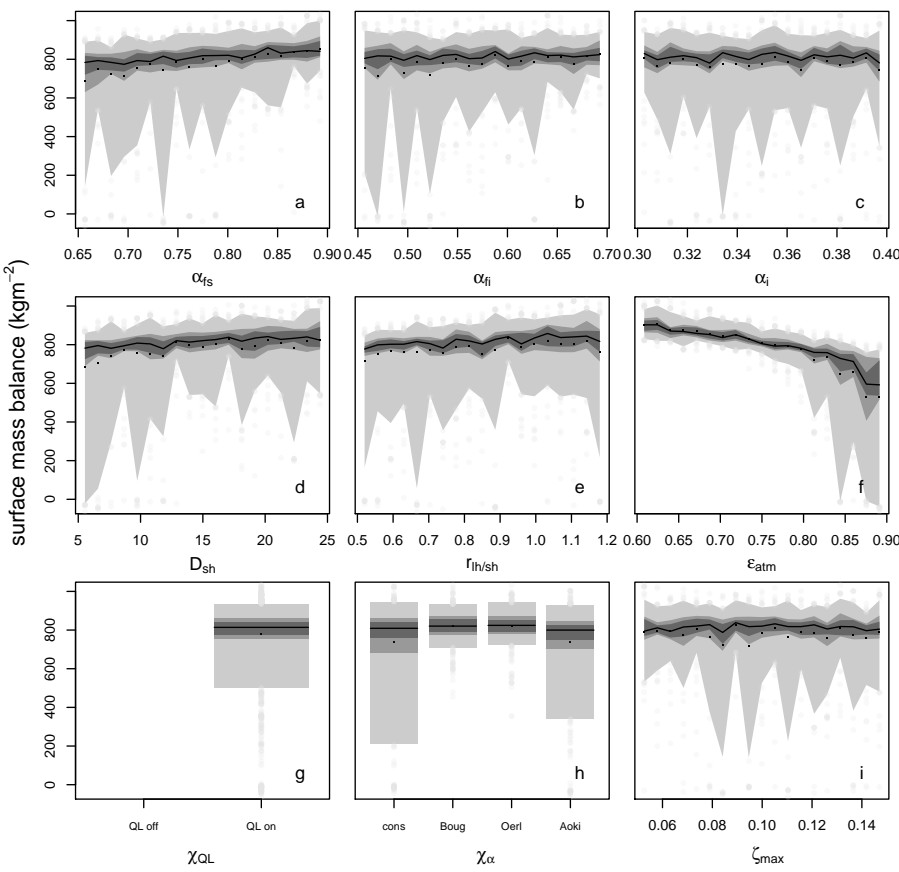

**Figure 7.** The ensemble statistics for the surface mass balance of the $Q_{Lon}$ sub-ensemble for region 5 (west $1000 - 2000$ m) at LGM. The 5/95, 25/75, 33/66 and 50 quantiles are shown in progressively darker shading. Black points represent the ensemble mean and the grey points correspond to the rest of the ensemble, apart from outliers (max five per bin allowed), which are removed to improve readability. Each plot represents the range of one parameter with $\alpha_{fs}$, $\alpha_{fi}$ and $\alpha_i$ in the top row, $D_{SH}$, $r_{lh/sh}$ and $\epsilon_{atm}$ in the middle and $Q_{Lon/off}$, $\chi_\alpha$ and $\zeta_{max}$ at the bottom. As $Q_{Lon/off}$ and $\chi_\alpha$ have two and four discrete values, the parameter range is not split in 20 intervals. Figure A.A2 shows a similar plot for the entire LGM ensemble.

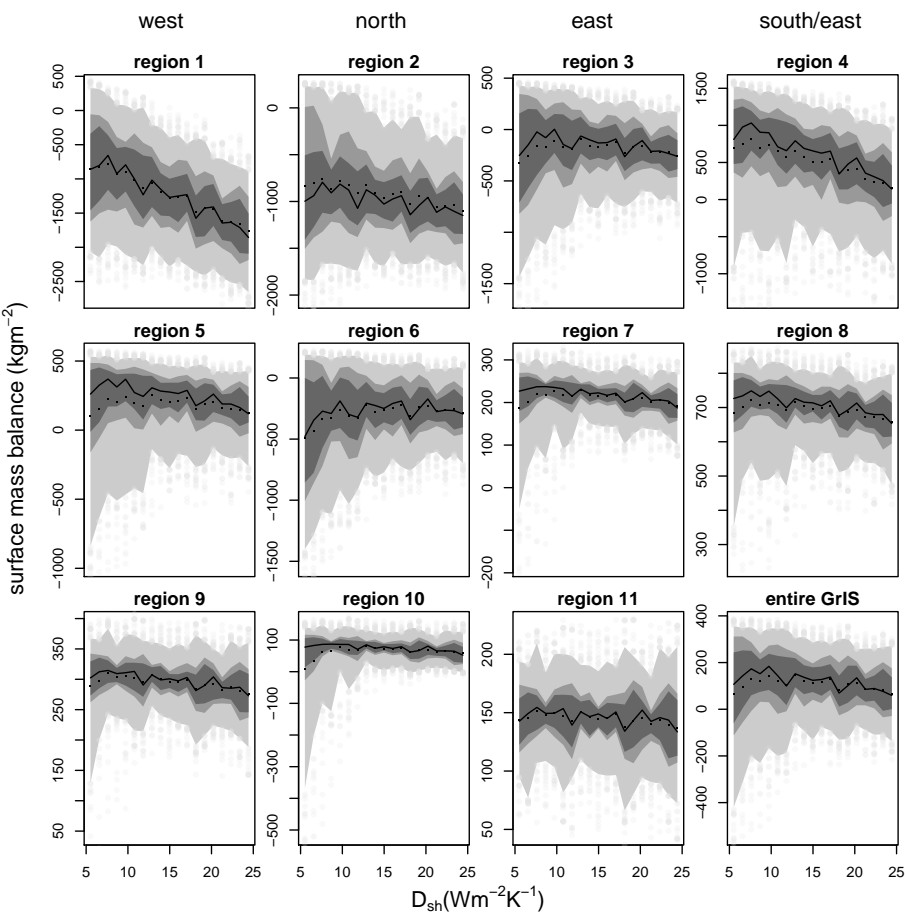

**Figure 8.** The dependency of the SMB on the turbulent heat exchange coefficient $D_{SH}$ is displayed in a similar matter as Figure 8 for PD. The shading represents the different quantiles: 5/95(light grey), 25/75 (grey) 33/66 (dark) and the median (solid black line). The black dots are the ensemble mean based on 20 intervals. The panels are sorted by elevation and regions, with the lowest elevation (0-1000 m) in the top row, and the highest elevation at the bottom. Each column is related to a region in Greenland: W-N-E-SE/S (fig. 5).

coefficient $D_{SH}$. The turbulent sensible heat flux varies over the different regions (fig. 8), which is why we present a deeper look on this particular parameter. Only the sub-ensemble with an active turbulent latent heat flux ($\chi_{QL}on$) is shown in Figure 8 because in regions without melt 50% of the simulations ($\chi_{QL} - off$ sub-ensemble) will show a similar mass balance. The overall width of the SMB distribution decreases with altitude. In the lowermost regions in the west and southeast (1,4) there

is a trend to more negative mass balances with larger exchange coefficients. Regions 3, 5, 6, 10 and to a lesser extent 2 and 7 show a distinct decrease in surface mass balance spread of the ensemble with increasing $D_{SH}$ and slight decrease of the mean (excluding region 6). The general slightly negative trend for the SMB with $D_{SH}$ is a result of higher air-temperatures than snow surface temperatures and a net heating of the turbulent latent heat flux. The negative trend is most pronounced at the lower regions of the west and south, where warm air advection is frequent, resulting in increased melt due to the heat supplied

by the sensible heat flux. The regions are also quite moist, in particular the south east (region 4), leading to a positive turbulent latent heat flux which in turn heats the surface even more.

The north-east (region 2, 3, 6, 7) of Greenland is colder and drier resulting in decreased turbulent fluxes and therefore the effect of $D_{SH}$ on the SMB and surface energy balance is less. The ensemble spread is narrower at higher altitudes (7-11) due to a generally positive mass balance, so an increased energy input due to any parameter will mainly impact the snow

temperature, but in the absence of melt the mass balance does not change significantly. Still the temperature change alters sublimation accounting for the remaining variability. This is pronounced in region 10, at low exchange coefficients melt is still possible, if other parameters result in a strong positive energy input, leading to a skewness towards negative SMB. At a stronger exchange the tight coupling with the heat reservoir of the atmosphere limits melt and the skewness towards negative SMBs vanishes, while the median stays almost constant. The parameter $D_{SH}$ is not the main driver for the SMB in region 10,

rather the parameter acts as a buffer of the SMB. At strong turbulent sensible heat exchange the surface temperature will be buffered by the air-temperature heat reservoir. The buffering effect is visible for most of the regions, where air-temperatures most of the year below the melting point (2,3, 5, 7-10).

The earlier discussion mainly focused on the impact of $D_{SH}$ on the surface mass balance via the turbulent sensible heat flux $Q_{SH}$, but $D_{SH}$ also impacts the turbulent latent heat flux ($Q_L$ eq. 12, additional figure in the Supplement). $Q_L$ gets more

negative with increasing exchange coefficient. There are two effects present, the increased surface water vapor pressure due to higher surface temperature as a result of $Q_{SH}$ and the actual water vapor exchange rate. The annual average of the ensemble over Greenland is a negative turbulent latent heat flux, meaning sublimation occurs more often than condensation. $r_{lh/sh}$ does increase $Q_L$ in its absolute value and therefore leads to more sublimation, but the effect on the mean is also lower than on the whole ensemble. At larger exchange rates more mass can be moved and therefore the variation over the ensemble increases as

larger surface temperatures are prevailing..

The average SMB increases at low elevation if $\chi_{QL} - on$, and decreases above 2000 m and is almost constant above 3000 m, where the annual average of the latent heat flux is almost zero. The first is a result of reduced melt due to a negative latent heat flux (sublimation), and the SMB increase is therefore most pronounced in the north-east. The latter is a region where melt only occurs for a couple of extreme ensemble members, so the sublimation is the only mass loss and therefore the SMB decreases

with increased sublimation via increasing $D_{SH}$.

**Ensemble statistics of other parameters for PD:** The other parameter results are consistent with the GSA: An increase in $\epsilon_{atm}$ decreases the SMB, but melting increases aproportionally with higher emissivity. The high impact of $\epsilon_{atm}$ is mainly related to the all year around impact altering ice exposure and albedo decay. The snow albedos increase the SMB, with the fresh snow albedo being more important. The SMB ensemble plots do not show any dependency on ice albedo, liquid water content and the mean is also unaffected by $r_{lh/sh}$. Additional figures are available in the supplement.

**LGM** The general features are similar to PD. Larger values of $D_{SH}$ reduce the variability/impact of the other parameters, and resulting in slightly lower SMB. Similar to the GSA a shift to lower elevations is seen. The impact of $D_{SH}$ on the SMB is negligible above 2500 instead of above 3000 m for PD and melt tails are limited to below 2000 m. $\epsilon_{atm}$ has less influence during the colder period, as well as the snow albedo related parameters.

**Sensitivity of the 10 m firn temperature:** In addition to the SMB, the sensitivity of the 10 m firn temperature was assessed. This was not possible for the GSA due the limited sample size. We limited the analysis of the firn temperature to 13 locations in the interior of Greenland. The firn temperature at these mostly central locations is sensitive to three parameters: $\epsilon_{atm}$, $\alpha_{fs}$ and $D_{SH}$. The first two have a rather linear impact with increasing temperature with increasing emissivity and decreasing albedo. The turbulent heat exchange coefficient has a similar effect on the 10 m firn temperature as shown for region 5. At small $D_{SH}$ the 10 m temperature has a larger variability depending on the other two sensitive parameters, while at large values the air-temperature buffers the snow temperature, even down to 10 m. The sensitive parameters are the main drives of the SEB, which determines the 10 m temperature, in those areas as discussed previously for the SMB. The only difference to regions at higher elevation of the SMB is the insignificance of $\chi_{QL}$. The turbulent latent heat flux has only a minor importance for the SEB, but in the absence of melt the vapor flux is the only SMB change. Tuning the model for the 10 m firn temperature only provides information about the sensitive parameters, which are $\epsilon_{atm}$, $\alpha_{fs}$, and $D_{SH}$.

## 4   Discussion

In this study we assess the model sensitivity due to parametric uncertainties. Cloud cover and the associated atmospheric radiation are large uncertainties both, in present day and glacial climates. The uncertainty in cloud cover is represented by the large parameter uncertainty of the atmospheric emissivity $\epsilon_{atm}$. The SMB is most sensitive to $\epsilon_{atm}$ under PD conditions. The sensitivity of the SMB is not drastically different during the LGM. However, lower atmospheric temperatures reduce the impact of the atmospheric emissivity, while also leading to fewer areas where melt and runoff occur. The relative contribution of the mass flux associated with the turbulent latent heat flux to the SMB increases drastically during the LGM (4 to 15 % of the total mass flux), making the turbulent latent heat flux switch the model's most sensitive parameter in large parts of the ice sheet. The increased importance of $Q_L$ is due to the absence of melt, similar to the highest elevations during PD climate. Additionally, SMB values have a smaller magnitude, as precipitation is less during the LGM and therefore the relative contribution of the vapor fluxes to the SMB is larger. For an accurate modeling of the SMB over the glacial cycle an inclusion of the turbulent latent heat flux is necessary, which may not be as important for a warmer climate.

The sensitivity metric we applied is a relative measurement that depends on two components, the absolute strength of a particular flux and the chosen parameter uncertainty range of the parameter. The latter depends on the subjective choice. We based the parameter values on published common ranges (tab. 1). The incoming long-wave radiation is the largest single energy source for the surface energy balance during PD, ranging from twice the incoming solar radiation around the margin to about

1/3 in the interior of Greenland in the annual average. Therefore, the impact of the atmospheric emissivity is decreasing from the coast inwards for multiple reasons: First, temperatures are higher around the margin leading to increased incoming thermal radiation. Second, cloud cover is more and therefore solar radiation is reduced. Lastly, at negative SMB the rate of melting depends greatly on the total energy flux and albedo decays faster for a warmer snow pack if more long-wave radiation reaches the surface during winter.

It is important to differentiate between the sensitivity of the Greenland wide integrated mass balance and the local SMB, as well as the impact of the individual fluxes on the absolute SMB. The Greenland-wide surface mass balance is most sensitive to the atmospheric emissivity during PD (fig. 2), while during the LGM the SMB shows increased sensitivity to the fresh snow albedo, the choice of albedo parameterization, and the turbulent latent heat flux. At the LGM, lower air temperatures and an ice sheet with less melt overall increases the importance of the vapor fluxes similar to high elevations at PD. It is therefore

not a necessity for models to include $Q_L$ in a warm climate, though desirable. Conversely, the turbulent latent heat flux $Q_L$ cannot be ignored during a colder and dryer climate. In addition, this means that although the cloud cover uncertainty may be similar during the colder period of the LGM the sensitivity of our model towards the emissivity uncertainty is less. Simple surface mass balance models like enhanced temperature index models are likely to create a bias as solar insolation changes are accounted for while the impact of the cloud cover on other components of the energy balance are neglected or implicitly

included in the PDD factor, this is even more true as even under PD conditions the impact of clouds on the SMB is highly variable (Van Tricht et al., 2016; Hofer et al., 2017).

At the local scale such as in the interior of Greenland even in PD climate the vapor flux ($Q_L$) is up to 1/3 of the total SMB, despite its small impact on the Greenland wide integrated scale. In addition, in the absence of melt the sublimation and condensation are the only changes to the SMB with a fixed precipitation forcing. Neglecting sublimation and condensation will

result in fundamental biases over long-term simulations of the Greenland ice sheet, in particular in its interior. This needs to be considered when tuning surface mass balance models for long time scales. A tuning for the Greenland wide SMB will mainly constrain the most sensitive parameters, which constrain certain key regions of high mass turnover, but not for the bulk surface. Furthermore, the temporal differences of the sensitivities on the local scale indicates that models of reduced complexity may fail drastically for other time periods (absence of $Q_L$ for example).

The sensitivity analysis shows that the uncertainty of the long-wave radiation has a larger impact on the SMB uncertainty than the uncertainty in the incoming solar radiation, but as it is not defined relative to the absolute flux, it does not necessarily tell us that the SMB is most sensitive to the long-wave radiation energy component. The impact each energy flux has on the absolute SMB has to be analysed separately. The long-wave radiation dominates, followed by the solar radiation. The larger sensitivity of the $\chi_{QL}$ switch at the LGM is mainly due to its increase on the absolute SMB. In the absence of melt, and with

reduced precipitation sublimation accounts for a larger portion of the absolute SMB.

It is beyond the scope of BESSI to resolve all the physical processes. We use a simple parameterization for the incoming long-wave radiation which does not accurately represent reality. The atmospheric emissivity is neither constant in space nor time. Area distributed values may work during the observational period, but differences in the atmospheric circulation alter these patterns over the glacial cycle. We conclude that the overall model uncertainty can effectively be improved by changing the simplified representation of the long-wave radiation flux as a function of atmospheric temperature and emissivity for either a more sophisticated parameterization or to use long-wave radiation as climate model input. If the model is to be tuned for the Greenland-wide SMB it will be biased towards the melt regions around the margin and therefore the atmospheric emissivity. Where possible parameters should be calibrated via quantities which they are sensitive for. Neither of our parameters are to be assumed constant in space, as albedo for example strongly depends on impurities and snow temperature, but unless the uncertainty of the incoming long-wave radiation is reduced, it is justifiable to work with optimized values from PD. The current model version does not use wind fields, although they impact the SEB via the turbulent fluxes. The strength of the turbulent exchange does not have a large impact on the SMB and the approach to neglect wind speed variability is therefore justified in the context of more uncertain parameters. The found model sensitivity towards the parameters is to be set into context with the assumed forcing (climate) uncertainty.

## 5   Summary and conclusions

The surface mass and energy balance model BESSI has been improved by accounting for turbulent latent heat flux and snow aging. The sensitivity of the model to the new implementations and uncertain model parameters was assessed with a variance based sensitivity method based on two ensembles with a total of 16500 simulations. Warm present day and the cold last glacial maximum climate were used to study the differences of the model response under different PD and LGM boundary conditions. The sensitivity analysis reveals that the inclusion of the turbulent latent heat flux is a necessity to simulate the local SMB and the integrated SMB over the entire Greenland ice sheet. The relative importance of sublimation and condensation is larger in the dry and cold climate of the LGM, as air temperature and precipitation are lower.

The uncertainty associated with cloud cover and atmospheric emissivity dominates the SMB model uncertainty. With the different circulation during the last glacial a changing energy input from the atmosphere to the surface will result in a SMB response. The sensitivity study further reveals that the uncertainty of the SMB as a result of the atmospheric radiation decreases in colder climate.

We find that uncertainties in the ice albedo, liquid water content and differences of the turbulent fluxes are of minor importance for our and likely also similar models. In order to reduce model uncertainty most effectively, first the larger energy sources of short wave and long wave radiation need to be constrained via the snow albedo and the atmospheric emissivity.

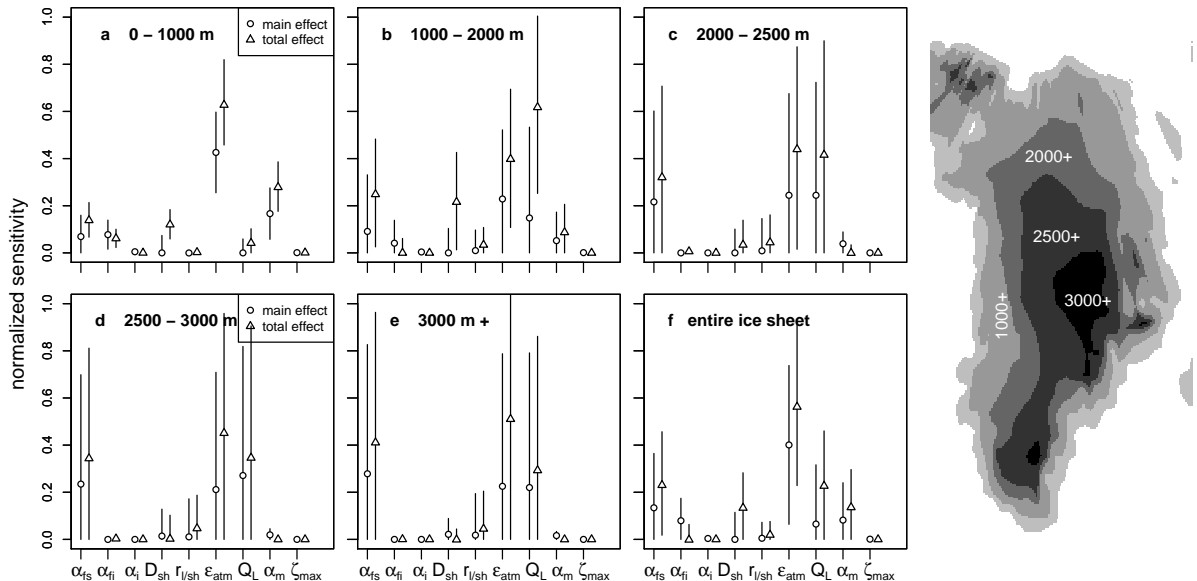

**Figure A1.** The global sensitivity analysis of the LGM-SMB provides the main order effect (circle) and the total effect (triangle). The two indices are displayed for all nine parameters over the different elevation bands ranging from 0-1000, 1000-2000, 2000-2500, 2500-3000, above 3000 m and the entire ice sheet. The symbol represents the mean value of the sensitivity index with the bars as $\pm\ 1\sigma$. The elevation bands of the LGM topography over Greenland are displayed on the right, the analysis is only done for cells where ice is present. The uncertainty of the sensitivity indices is larger than for PD due to the smaller ensemble size.

*Code availability.* The BESSI model code is available on git-hub (https://github.com/TobiasZo/BESSI/tree/TobiasZo---GSA-model-version). Additionally, the github branch also contains the analysis and plotting scripts. It is also available together with the supplement at 10.5281/zenodo.4310369.

## Appendix A: Additional plots

*Author contributions.* TZ implemented the model changes, conducted the ensemble simulations, the sensitivity studies and the data analysis and wrote the main part of the manuscript. AB contributed to all aspects of this study.

*Competing interests.* The authors declare that they have no conflict of interest.

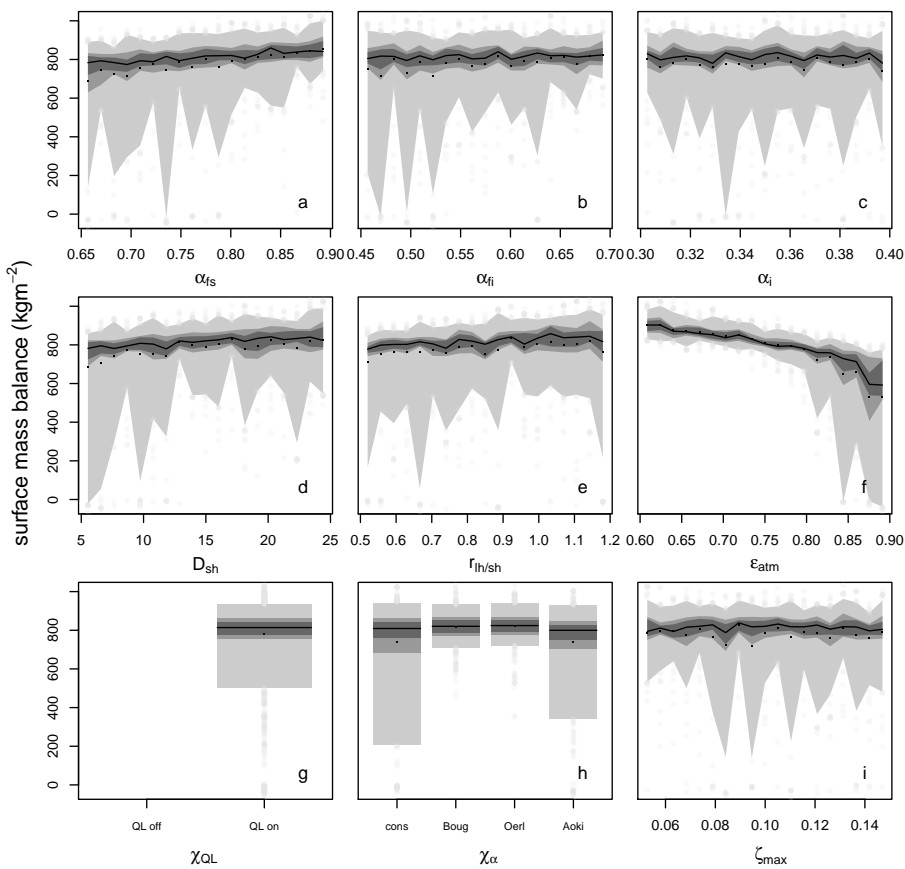

**Figure A2.** The ensemble statistics for the surface mass balance for the entire ensemble at region 5 (west 1000-2000 m) at LGM shows the 5/95, 25/75, 33/66 and 50 quantiles in progressively darker shading. Black points represent the ensemble mean and the grey points correspond to the rest of the ensemble, apart from outliers (max 5 per bin allowed), which are removed to improve readability. Each plot represents the range of one parameter with $\alpha_{fs}$, $\alpha_{fi}$ and $\alpha_i$ in the top row, $D_{SH}$, $r_{lh/sh}$ and $\epsilon_{atm}$ in the middle and $Q_{Lon/off}$, $\chi_\alpha$ and $\zeta_{max}$ at the bottom. As $Q_{Lon/off}$ and $\chi_\alpha$ have two and four discrete values, the parameter range is not split in 20 intervals.

# Sensitivity of the turbulent latent heat flux in PD

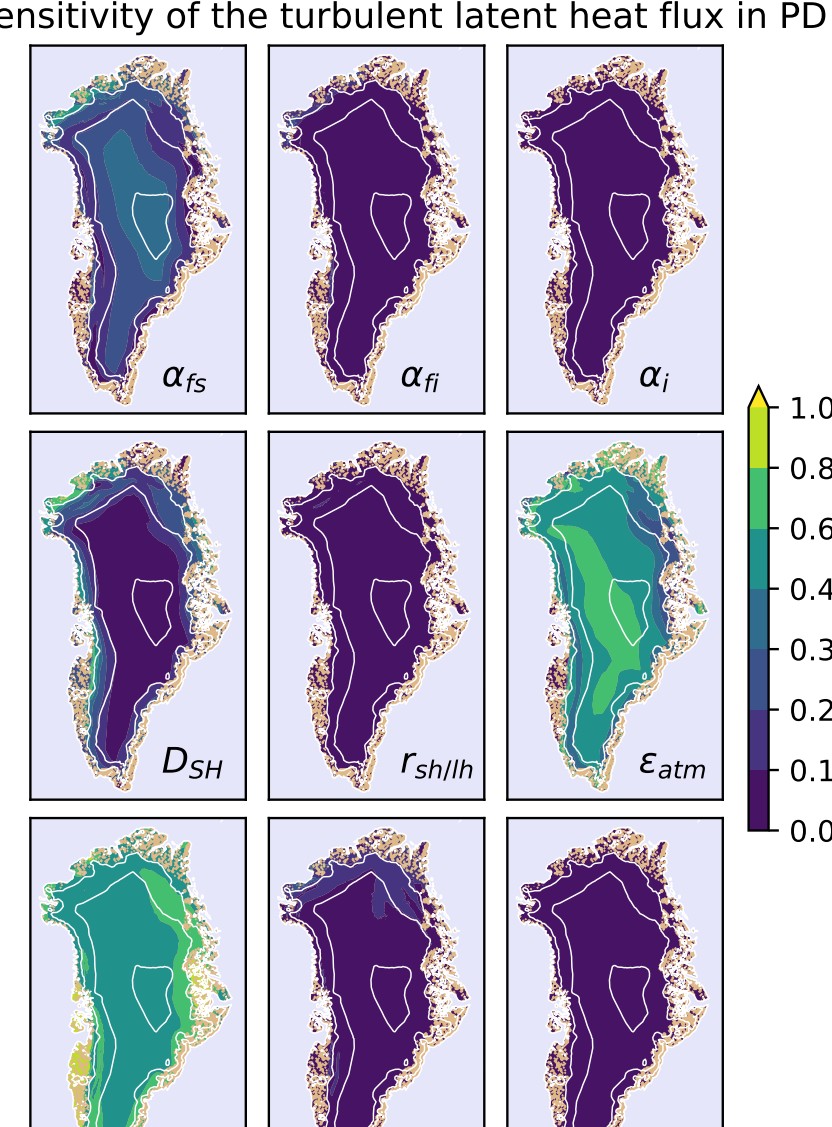

**Figure A3.** Global sensitivity of the turbulent latent heat flux at PD. The total sensitivity index of $Q_L$ of every parameter for PD is displayed for every ice covered grid cell. The ice free land is in brown, the ocean in blue.

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
