# Peer review of "Sensitivity of the Greenland surface mass and energy balance to uncertainties in key model parameters"

_The Cryosphere, 2019_

## Referee Comment (RC1) · Anonymous Referee #1 · 19 Jan 2020

**Summary**

The paper evaluates the sensitivity of output from a simple glacier surface mass and energy balance model to changes in its parameters. For this study the model has been extended from a previous version to include the description of turbulent latent heat fluxes, deemed important for cold climates. The evaluation is performed over the Greenland ice sheet for two contrasting climate states (present day:PD and last glacial maximum: LGM) and several regions with distinct surface mass balance regimes. The work provides detailed information about the importance of key model parameters for the surface mass and energy balance and confirms the importance of latent heat fluxes

for the LGM climate.

**General comments**

The conclusions of the paper are fairly specific for this particular model. I was wondering if the manuscript wouldn't find a more appropriate audience if it was instead submitted e.g. as a model evaluation paper in GMD. This is an editorial decision, but I think it is worth considering.

The paper is well organised and the text is largely clear in its presentation in sections 1-2, which requires some improvement to be more precise (see detailed comments below).

My main problem with the results section (Sec. 3) is related to the challenge to present results for 9 different parameters for two different climate states, 6-12 different regions and two different analysis techniques in a concise and interesting way. I believe this section in its present form lacks focus and direction and much improvement can be made in presenting these results. More precision of the descriptions is also needed here.

I suggest the authors should look for possible generalisations across these four axes of analysis and of clear story lines that are followed throughout the discussion of results. To give an example: the manuscript discusses the notion that the Greenland margins at the LGM exhibit similar features to the interior at the PD. Maybe this can be used to generalise the results further and reduce the amount of individual cases that need to be discussed.

Overall, one possible approach could be to define a few main conclusions of the study first and then find evidence for those in the different results. If possible, consider moving less important results to an appendix or supplement.

What struck me as particularly difficult to digest are parts of the text where the description of the results happens without accompanying figures (e.g. p17.l21, p19.l17).
I strongly suggest to provide additional figures to make the discussion of results more tangible. In all cases where results are not shown in figures or tables, add 'not shown' in the text, otherwise, make a reference to the figure.

Regions. It is confusing to me that the paper apparently operates with three different sets of regions (those defined in Fig2, Fig 5 and those not shown for the LGM). If at all possible, I'd suggest that one set of regions (or at least one clear definition of regions that may then results in differences between PD and LGM) be used throughout the manuscript? In the current framework, the regions defined for the LGM should be shown (they should be different from the PD if they are based on elevation) as results are given for those. It seems that the PD analysis is limited to region 5 only, so it is a surprising choice to show all PD regions in detail, but none of the LGM regions.

As pointed out in the text, the relative importance of different parameters in the global sensitivity analysis are dependent on the sampled parameter ranges. I miss a clear motivation and argument for the plausibility of the assumed ranges beyond reference to Born et al. (2019). This seems like an important aspect of the paper, so it should get some attention in the text. This includes the question if the parameter ranges supposedly derived for PD climate hold for the LGM and what could be done to mitigate this effect, if any.

**Specific comments**

Up front a few points that are repeated throughout the manuscript. Some examples are given below.

- Reconsider the use of 'it' and 'this' in cases where the subject of the sentence is not clear.

- Distinguish between physical quantities and mechanisms on one hand and the parameters that influence those on the other hand.

- Be clear about what is shown in a given figure (PD or LGM) and what results you are

discussing in a given paragraph (PD vs LGM). Possibly use section headers to make that distinction.

- Be explicit what you are comparing in a given part of the text: PD vs. LGM, one region against another or one parameter against another.

- In the context of this paper, I would always refer to the contrast between PD and LGM as 'difference' rather than 'change', since there is no time dependence here.

Title: Add 'surface' before 'mass and energy balance'

Abstract: Could add more information about the model and experimental setup.

p1.l2 remove 'climate' after 'present day'. A climate is not a period.

p1.l16 If this is supposed to be a reference for the ITM method, more appropriate references may be Bintanja et al., 2002 or Van den Berg et al., 2008

p2.l2 Add 'computationally' before 'too expensive'.

p2.l9 Maybe 'can be used', to make clear they are not used in parallel.

p3.l9 Not obvious what a 'mass following' grid is. Clarify, add a reference.

p3.l9 Add 'in the snowpack' after '15 layers' if that is the correct description.

p3.l9 Maybe 'The mass of each layer is 100 - 500 kgm-2'. Clarify if the mass is decreasing/increasing with depth or where the range originates from.

p3.l12 Clarify what happens to the other variables if they are not downscaled to the model topography. Are they just interpolated?

p3.l17 Which two? The last two? Clarify

p3.l24 Insert 'then' after 'The actual melt is'.

p4.l1 Consider adding numbers for the 4 different parametrizations below. 1. Constant, 2. Oerlemans and Krapp ...

p4.l12 We don't know yet how large the boxes are! Also, you use the term 'layers' before. Is large the right term for a layer defined by its mass?

p4.l12 The end of the sentence 'would likely already be wet if the real surface was resolved' needs further explanation to be comprehensible.

p4.l14 Consider using same notation as for the other forms (number) author: ...

p4.equ(4) Add 5d for Ts = 273K?

p4,l21 Clarify what the first 'this' refers to in 'this was adapted for this model'.

p4.l24 Clarify what 'fixed and temperature dependent' means. How can it be both at the same time?

p4.l25 Clarify what 'it' refers to in 'Keeping it constant'

p5.l7 Could explain physically what this approach is trying to mimic. Supposedly that the first snow that falls on a wet surface will get wet immediately.

p5.l21 'ice-sheet' –> 'ice sheet'

p6.l16 Typo 'fesetup' –> setup

p6.l23 'heat supplied by precipitation depends on the temperature'. Which temperature?

p6.l24 Insert 'in the snowpack' after 'grid cell'. Consider consistent use of terms 'layer' vs. 'grid cell' vs. 'grid box'.

p7.l12 'The global sensitivity analysis is a variance-based method'

p7.l13 'In contrast to other methods'

p7.l13 'all parameters are varied at the same time'

p7.l15 remove 'using' after 'hypercube'.

p7.fig1 Why is the colour over the ocean different in a and c? Why is there a contour line in the ocean? Are elevations at LGM relative to LGM sea level (as they should) or relative to PD sea-level? Suggest to plot all topographies with ocean surface at sea-level = 0.

p8.l8 'optimal model setting' implies an already tuned model which is supposedly not the case here. Clarify!

p8.l18 Specify temperatures in degree C instead.

p8.l19 'As the ice sheet has different shapes' –> 'As the ice sheet geometry differs between the two climate states' or similar.

p8.l19 Why not use the larger ice sheet area of the LGM for both to avoid biases e.g. from the large ocean area in the south-east?

p8.l25 How is the model run back and forth? Is it reversible in time?

p8.l31 Explain why the ensemble is split in two.

p9.l1 Suggest to move 'A detailed description of the algorithm can be found in (Sobol et al., 2007) and (Saltelli et al., 2010).' after 'used to estimate the model sensitivity.' to where the general description of the method ends. Also, specify k in your example.

p9.l15 Insert 'STi' after 'total sensitivity index'.

p9.l16 Clarify temporal or spatial average in 'surface mass balance and the average surface temperature'.

p9.l17 Specify which variables are averaged and which are summed.

p9.18 'reseeding' –> 'residing'

p10.fig2 Do you explain somewhere how the sensitivity is normalised? The plot of Greenland looks distorted. Can this be plotted in equal aspect ratio. Additional (white) contour lines could make the region separation clearer. Caption: Add panel for 'entire

ice sheet' to the description. Consider adding panel indicators a-f for the boxes and g for the GrIS plot and use them in the text.

P10.l4 Maybe 'low' –> 'limited'

p10.l4 Remove 'changes' after 'SMB'

p10.l5 See general comment on relative sensitivity of parameters dependent on choice of parameter range. Should be clarified here.

p10.l6 Check consistency Eatm (here) - Eair (Figure 2)

p10.l7 Check consistency DSH (here) - Dsf (Figure 2)

p10.l9 Add 'albedo' after 'the fresh snow'

p10.l14 is 'above 2000 m' > 2000 or the region 2000-3000? Clarify

p11.fig3 Title: 'Sensitivity of the SMB at PD' Are values >1 actually defined for this analysis? If not, why is the yellow arrow on the colour bar? What does a sensitivity index above 1 mean? The colour scale with the darkest colour for the least important results is not convincing me. Contour lines are not visible on most plots. Suggest a different (lighter) colour and omitting the numbers. Caption: Mention figure is for PD. Mention colour choice for ice free land. 'mass balance' –> 'surface mass balance'. Reformulate 'are not to be taken too seriously'. If we don't take the absolute values seriously, what is left in this figure? If relative values are more important, find a way to plot those instead.

p12.l2 Reformulate 'the structure is much more complex'. What does that mean? There is more information in 2D compared to 0D?

p12.l3 remove 'glaciers' after 'west coast'.

p12.l7,8 2x 'becomes' –> 'is'. Otherwise this implies a process.

p12.l30 Reformulate 'negative ensemble member'.

p12.l33 'interior of Greenland'

p12.l33 Clarify 'the atmosphere is more in balance with the snow surface'. What does that mean?

p12.l34 Consider adding a section header 'LGM analysis' or similar to make a clearer separation between the two climate states in the text.

p14.l8 Would an additional figure similar to Fig2 for the LGM help?

p13.fig4 Title: 'Sensitivity of the SMB at LGM' See comments for fig3 for general layout. Caption: Start with description what is shown, not with discussion of the figure.

p14.l2 'The ice sheet integrated SMB'. Do we see this somewhere? Consider adding a figure or table and refer to it here.

p14.l2 'shows sensitivity', add qualifying statement 'some', 'strong'. Depending on relative scale, all most parameters will show some sensitivity.

p14.l4 'do not impact the SMB'. Not at all? Clarify.

p14.l4 add 'the' in 'either of the two climate states'. Refer to fig 3 and 4 then.

p14.l4-7 First start discussion of LGM, then continue comparison LGM-PD.

p14.l8 'increased'. Relative to PD? Clarify.

p14.l9 Add 'surface' before 'mass balance'.

p14.l11 'intra-annual variability' –> 'seasonal variability'

p14.l12-14 'the ... impact of ... impacts the latent heat flux more than the actual exchange coefficient'?? Not clear. Reformulate. Clarify.

p14.l15 remove 'temperature' after 'surface and the air', to avoid duplication.

p14.l16 'the fewest precipitation amounts'. Is this shown somewhere? If not, add 'not shown' in the text.

p14.l18 'more important on the western than the eastern margin'. Explain why.

p14.l18 Add 'at LGM' after 'The reduced model sensitivity'.

p14.l21 'The sensitivity of ...' to what? maybe 'model parameters'?

p14.l22 'without additional figures'. Not sure this is a good idea, see general point. Maybe add a table with results instead?

p14.l23 What is 'final firn albedo'? Reformulate

p14.l24-25 Not sure I understand this conclusion. Reformulate?

p14.l26 What does 'it' refer to in 'it is most sensitive'? Clarify

p14.l28 'as are the snow albedo related ones in the north' –> 'as are the ones related to snow albedo in the north'

p14.l30 Start sentence with 'Globally' and then give specific details in the end.

p14.l31 'the framework' , maybe 'this framework', 'our framework'

p15.fig5 Add panel with regions for LGM, which are different and actually used in the analysis (unlike only region 5 for PD). Why invent new regions after what is already introduced in figure 2? Could you not do the analysis for region 1000-2000 instead of region 5? Or find a common subset to use in fig 2? Caption: Start caption with what is displayed in the figure. Results are for the text.

p15.l1 'more' or 'less' than what? Maybe 'closely linked to the SMB'?

p15.l1 'and shows similar sensitivities as have been reported for the SMB'

p15.l2 'much lower.' compared to what? Why is that expected? Explain.

p15.l4 Add 'surface' before 'mass balance'.

p15.l6 'Parameters which result in either surface heating or cooling'. More precision needed. It is not the parameters that result in heating or cooling, but the physical

process that is parameterised.

p15.l7 'require a different analysis'. So what is this analysis, describe. Consider adding this description as 2.4.

p15.l7 Add 'Conversely' before 'Albedo', as these are examples where GSA will work.

p15.l8 Add 'so that GSA gives clear results' or similar after 'mass balance'.

p15.l9 'and 13 for the LGM (2 more around Elsmere Island)'. Need to show these in a figure.

p15.l19 'based on elevation and similarity'. Similarity of what? Explain.

p16.fig6 Add yticks in column 2 and 3. Suggest to plot all results as discrete boxes. The mix between continuous quantiles and discrete outliers looks strange to me. Caption: Add that this is for PD.

p16.l1 What does 'It' refer to in 'It shows'?

p16.l4-6 According to you, parameter Eatm can be well analysed with GSA. Why does it need more detail here?

p16.l6 'accelerating manner' could suggest that the parameter should be sampled non-linearly.

p16.l7 'the variation of the SMB as a result of other parameters, increases too' and p16.l8 ' the width of the distribution decreases' Could you explain why this is the case? I would think with more available energy (l7), differences in the other parameters have a larger impact on the SMB. Similar with lower albedo (l8), differences in the other parameters are more effective in making a difference.

p16.l8 Add parameter in 'Even very low albedo parameter values'

p17.l4 'has the highest median mass balance'. Explain why.

p17.l6 'The strong impact' on what?

p17.l7 'the other fluxes'. What kind of fluxes? Energy?

p17.l21 'During the LGM the western region between 1000 - 2000 m'. Should be shown.

p17.23 'three distinct changes' –> 'three distinct differences relative to PD'

p17.l23 '$\chi$QL results in a decrease of SMB'. Distinguish physical process and parameters.

p17.l26 'slower snow albedo decay'. This is discussed as a general result, but is not available in all albedo models, is it?

p17.l30- Make clear that the discussion is back to LGM results.

p17.l28 'sublimation ... results in a mass loss rather than a mass gain as in PD.' You should probably distinguish the opposite sign using the term 'deposition' or 'desublimation'.

p18.fig7 Panels are difficult to compare due to different vertical scale. Add xticks in row 2 and 3 Caption: Add that this is for LGM. Highest elevation at bottom is counterintuitive, consider changing order. Add figure with regions and link from here.

p19.l2 'The smallest spread of the ensemble is found in the high altitude-regions 9, 10, 11'. Difficult to judge with different vertical scales in figure. Also, this is a new point. First finish discussion of DSH?

p19.l3 'a result of higher air-temperatures than snow surface temperatures'. Needs more explanation to be clear.

p19.l4 'This shows'. What does 'This' refer to?

P19.l7 What is 'moisture differences between surface and atmosphere'?

p19.l8 '7-11' was 9-11 in the explanation above at l2. Clarify.

p19.l14 'it acts as a buffer of the SMB'. Not clear.

p19.l18 Add 'Ql' after latent heat flux.

p19.l17- Hard to follow this part without any guidance. Include a figure?

p19.l28 'the SMB decreases' with what?

p19.l29 'behavior to the GSA' –> 'behavior as shown in the GSA'

p19.l35 'Similar to the GSA' –> 'Similar to the results from the GSA'

p20.l1 'above 2500 instead of above 3000 m'. Is this comparing to PD? If so, mention it.

p20.l6 'in region 5'. And in in all the other regions? No discussion of those?

p20.l8 'the air-temperature buffers the snow temperature'. Not clear. Clarify.

p20.l14 'The model sensitivity in this study is evaluated' –> 'The model sensitivity is evaluated in this study'

p20.l15 'big' –> 'large'

p20.l17-18 I would say it is the other way around: lower atmospheric temperatures ... 'leading to fewer areas where melt and runoff occurs' and consequently 'reduce the impact of QLWin and $\varepsilon$atm' ...

p20.l20 'This is due to the absence of melt'. I read this as complete absence of melt. Is that correct? Otherwise 'negligible amount of melt'?

p20.l25 'bigger' –> 'larger'

p20.l25 'for the surface during PD' –> 'for the surface energy balance at PD'

p20.l26 Is it correct that the long-wave radiation is twice as large as the incoming solar radiation? This is for heavy cloud cover?

p21.l1 Add 'surface' before 'mass balance'.

[Figure]

p21.l12 'the temporal change of the sensitivities'. What is that? The contrast between LGM and PD?

p21.l15- Try to formulate this positively to make it clearer.

p21.l18 Replace 'increased' by 'larger'

p21.l19 'during the LGM' –> 'at the LGM'. Similar for PD and in other places.

p21.l19 replace 'increase' by 'influence'.

p22.l4 Move 'climate' to after 'glacial maximum' in the next line.

p22.l5 'study the change of the model response under different boundary conditions' –> 'study the differences of the model response under LGM and PD boundary conditions'.

p22.l6 add 'for the LGM climate' after 'is a necessity'

p22.l9 'creates a SMB model uncertainty' –> 'govern the SMB model uncertainty'

p22.l9-10 'With the change in circulation during the last glacial a changing energy input from the atmosphere to the surface will result in a SMB response' –> 'With the different circulation during the last glacial maximum a changing energy input from the atmosphere to the surface will result in a SMB difference.'

p22.l11-14 I don't think this has been shown in the manuscript. This could be a discussion item, but not a conclusion of the manuscript if it is not even mentioned before.

p22.l16-18 This reads like a discussion item, not like a conclusion.

References

Revise inconsistent use of abbreviated and full journal titles. E.g. Cryosphere –> The Cryosphere. J. Geophys. Res. –> Journal of Geophysical Research.

Bintanja, R., van de Wal, R. S. W., and Oerlemans, J.: Global ice volume variations through the last glacial cycle simulated by a 3-D ice-dynamical model, Quat. Int., 95-

96, 11-23, 2002.

van den Berg, J., van de Wal, R., and Oerlemans, H.: A mass balance model for the Eurasian Ice Sheet for the last 120,000Âăyears, Global Planet. Change, 61, 194-208, doi:https://doi.org/10.1016/j.gloplacha.2007.08.015, 2008.
* * *

---

## Referee Comment (RC2) · Andy Aschwanden (Referee) · 26 Feb 2020

Review of Zolles and Born

Characterizing model sensitivities is an important step towards reducing uncertainties in model projections, making the manuscript both timely and useful. The model setup based on Sobol Sequences is carefully done and the use of global sensitivity analysis appropriate, GSA should become a mainstay in a modeler's toolbox. My main criticism relates to the structure and langue of the paper, not the science. The manuscript lacks clarity and precision. To improve readability and to make the manuscript more engaging and flow better, I suggest some reorganization and rephrasing:

[Figure]

- When possible, switch from passive to active language (e.g. "Here we set x = 20" instead of "x=20 is being used in this setup"). "Figure Z shows what X depends on Y" could be rephrased to "We find that X strongly depends on Y (Fig Z)".

- When presenting and discussing results, make sure that figures and tables are referenced whenever applicable. This makes it easier for the reader to follow.

- Try to reduce the frequent use of "this" and "these" to improve clarity and readability.

- I think the equations underlying the different parameterizations that are part of your uncertainty quantification are not crucial for the message of the paper. Consider moving the details to an appendix.

- Since you only analyze and discuss main effect Sobol indices, you could simplify the paper by removing mentions of total effect indices, unless you have a good reason to keep them. (A single sentence what total effects are and why you exclude them may by sufficient).

- Before submitting I recommend having the manuscript proof read by a native speaker to iron out remaining minor issues.

- I agree with the other reviewer that more focus and precision is needed in the results section.

To the best of my knowledge, a novelty of this manuscript is to present spatial patterns of the Sobol indices. Aschwanden et al (2019) and Bulthuis et al (2019) only showed Sobol indices of scalar quantities. It may be worth pointing this out.

Fig. 6 vs 7: In Fig 6, the mean is always lower than the median, which is not the case in Fig 7. What does this say about the underlying PDFs?

In equations I recommend using roman font for sub- and super-scripts if they do not describe variables, e.g. $Q_{\mathrm{in}}$. Use of $SMB$ and variables like $SW$ are always tricky. Writing $(1-\alpha) SW_{in}$ could be interpreted as (1-\alpha) * S

* W_{in}.

Rewrite paragraph about the global sensitivity analysis (p 7, l 12-20). I wonder if it would be better to first outline how you designed the ensemble and what method was used to draw from the parameter space, and then introduce the global sensitivity analysis that you use to analyze your ensemble.

Please find technical comments attached. I've tried to make suggestion how to rephrase sentences here and there, but these comments are not exhaustive.

Please also note the supplement to this comment:
https://www.the-cryosphere-discuss.net/tc-2019-251/tc-2019-251-RC2-supplement.pdf
* * *

---

## Author Comment (AC1) · 26 Apr 2020

We would first like to thank the Referee for their constructive criticism of our manuscript "Sensitivity of the Greenland mass and energy balance to uncertainties in key model parameters". Please find the full response in the attached supplement.

Please also note the supplement to this comment:
https://www.the-cryosphere-discuss.net/tc-2019-251/tc-2019-251-AC1-supplement.pdf

---

## Author Comment (AC2) · 26 Apr 2020

We would first like to thank the Referee for his constructive criticism of our manuscript "Sensitivity of the Greenland mass and energy balance to uncertainties in key model parameters".

Characterizing model sensitivities is an important step towards reducing uncertainties in model projections, making the manuscript both timely and useful. The model setup based on Sobol Sequences is carefully done and the use of global sensitivity analysis appropriate, GSA should become a mainstay in a modeler's toolbox. My main criticism relates to the structure and language of the paper, not the science. The manuscript

lacks clarity and precision. To improve readability and to make the manuscript more engaging and flow better, I suggest some reorganization and rephrasing:

- When possible, switch from passive to active language (e.g. "Here we set x = 20" instead of "x=20 is being used in this setup"). "Figure Z shows what X depends on Y" could be rephrased to "We find that X strongly depends on Y (Fig Z)". - When presenting and discussing results, make sure that figures and tables are refer- enced whenever applicable. This makes it easier for the reader to follow. - Try to reduce the frequent use of "this" and "these" to improve clarity and readability.

We will follow these suggestions where possible.

- I think the equations underlying the different parameterizations that are part of your uncertainty quantification are not crucial for the message of the paper. Consider moving the details to an appendix.

Due to the inherent importance of the parameters in the aim of our study, we want to keep the equations within the main manuscript text.

- Since you only analyze and discuss main effect Sobol indices, you could simplify the paper by removing mentions of total effect indices, unless you have a good reason to keep them. (A single sentence what total effects are and why you exclude them may by sufficient).

We are only using the total effect in our discussion and paper. Considering the discussion in our paper, we will add one clear sentence why the main effect is not used.

- Before submitting I recommend having the manuscript proof read by a native speaker to iron out remaining minor issues.

OK

- I agree with the other reviewer that more focus and precision is needed in the results

section.

A more clear structuring of the results and discussion will be applied. In particular we want to follow the suggestions of reviewer 1 in regard to LGM and PD subheadings.

To the best of my knowledge, a novelty of this manuscript is to present spatial patterns of the Sobol indices. Aschwanden et al (2019) and Bulthuis et While al (2019) only showed Sobol indices of scalar quantities. It may be worth pointing this out.

Thank you for pointing this out to us. This will be emphasized throughout the manuscript.

Fig. 6 vs 7: In Fig 6, the mean is always lower than the median, which is not the case in Fig 7. What does this say about the underlying PDFs?

Fig 6 shows a particular region (region 5), where melt occurs and therefore the PDF has a longer tail towards negative values resulting in a lower mean than median. In contrast, figure 7 shows the sensitivity of SMB to a single model parameter for different regions. Here, the PDFs do not all have negative tails.

In equations I recommend using roman font for sub- and super-scripts if they do not describe variables, e.g. $Q_{\text{in}}$. Use of $SMB$ and variables like $SW$ are always tricky. Writing $(1-\alpha)SW_{in}$ could be interpreted as $(1\text{-}\alpha) * S * W_{in}$.

We will look into this in all equations and try to optimize it following the journal's guidelines.

Rewrite paragraph about the global sensitivity analysis (p 7, l 12-20). I wonder if it would be better to first outline how you designed the ensemble and what method was used to draw from the parameter space, and then introduce the global sensitivity analysis that you use to analyze your ensemble.

We will rewrite the paragraph but would prefer to keep the order as it is now.

Please find technical comments attached. I've tried to make suggestion how to

rephrase sentences here and there, but these comments are not exhaustive.

We thank the referee for the detailed technical comments and will address a few specific ones here. We will submit a complete list of replies to all comments along with our revisions.

How do you downscale temperature, and how do you treat other variables:

All variables are interpolated bi-linearly to the horizontal model grid. Only the atmospheric temperature is corrected for the actual model topography using a temperature lapse rate of 0.65 for PD and 0.85 for the LGM.

Why 273K, not 273.15K for the melting point?

We are using 273.15 K in our model, we will change the manuscript and use the actual value or a corresponding symbol throughout the revised text.

Add a column with references to justify the range. Also consider listing the distributions used.

We will be adding references to Table 1. All parameters are distributed following a pseudo random sobol sequence.

Conclusion section: Here you introduce a new idea/model that has not yet been discussed.

The new idea introduced in the conclusion will be moved to the discussion.

---

## Referee Report (RR1)

**General comments**

This manuscript investigates the parameter sensitivity for a glacier surface mass and energy balance model. I'm less familiar with this model, which calculates energy balance using just temperature, precipitation, and incoming shortwave radiation as inputs, and parameterized other components, but the model is well-described. Authors perform experiments for parameter sensitivity over Greenland for two time periods: the present day and Last Glacial Maximum (LGM).

The introduction sets up the paper well. I think that the experiments in this paper, showing which parameters mass balance is most sensitive to, are important and will be useful for others simulating mass balance for glaciers and ice sheets. The presentation of results isn't always intuitive, and at times the manuscript does not clearly and accurately communicate the key messages.

**Specific comments**

p1 L4: Is 'invariant' the correct word? You show that the sensitivity varies by region (Fig 3).

P1 L4: Should you clarify that emissivity has, by far, the highest sensitivity (at least in present day)?

P1 abstract: Is it useful to note differences between present and LGM? The comparisons seem like a big part of the paper.

P1 L22: The use of the word 'unfortunate' here feels like an opinion. This occurs throughout the text, with words like 'rather' and 'clearly'. I think these words should be removed.

P9 Table 1: Does parameter #5 have a citation?

P9 Table 1: Parameter #8, I think a different abbreviation is used later in Fig 2 (am). Keep these consistent.

P10 L16: Lots of acronyms here. Better to say 'Present' instead of PD?

P10 L21: remove 'a slight influence of'

Fig 2: Great, this seems like a key figure. It's not clear what the difference between main and total effect are. Did I miss this in the methods? Also, as a key figure, why is the equivalent for LGM not in the main text? It seems like it would be more useful to have that figure in the text, and current Figures 1 and 5 either smaller or in the supplement.

P15 L6-7: Delete 'Due to their large size,' and 'but we include the main findings here'.

P15 L30: I'm confused here. Maybe just 'Conversely' needs to be removed?

P 16 L 5: Why pick region 5? Later you mention the ELA is within the region, maybe include that here.

Figure 6: Text says that a) and b) show sensitivity, but not c) (P16 L 10-14). But they all look pretty similar to me. Maybe labeling SMB on all axes would help.

Figure 6: Why are the lightest color vertical bars, while the 3 darker colors are continuous?

P18 L 22-25: Really hard to follow the list, adding numbers in front of each point could help (e.g. 1) The impact of …)

Figure 7: I'm a bit confused here, you're looking at the sensitivity of parameters on surface mass balance only with QL on, still for region 5 at the LGM? I would expect it to be the equivalent to Fig 6 but for LGM, but what is going on with QL?

Figure 8: This is figure 8, typo in caption?

Figure 8: Why did you choose to show parameter sensitivity by region only for Dsh?

Discussion: I found this section a bit hard to read. Some sentences might need commas to make the point clear.

P23 L12: delete 'while' and rewrite? There is still sensitivity to emissivity during the LGM.

P23 L15: what does 'though desirable' mean?

P24 L8: Change 'neither' to 'None', as you have multiple variables?

P24 L16: Do you show you improve the model with these new additions?

P24 L 23-25: Is there a previous description of the different circulation during the LGM? Having it just brought up here is missing an explanation if one doesn't know about it.

Appendix: Some plots mentioned in the text are missing, or is there also a supplement? If so, why are some in Appendix and some in supplement?

---

## Editor Decision (ED1)

Dear authors,

Thanks for submitting your revised manuscript. And I am sorry for not responding earlier, but I was ill with covid-19 in December so I didn't manage to handle your revised manuscript until now.

In the first round of reviews one of the referees was more critical that the other. Unfortunately, the more critical referee will not be able to assess the revised manuscript and I will therefore send it out for review by one additional (new) reviewer. Before I will do that, I have some requests for you.

Can you please submit a version of the manuscript in which all changes (wrt. your initial submission) are highlighted? As it is now, I find it quite difficult to track which changes you have made, and where. This will also greatly help a new reviewer with his/her work in assessing the revised manuscript.

I also encourage you to check if the things you write in your reply are correct. As an example you write: We want to avoid using a supplement, but will move content to the appendix. The need for additional figures is a shared concern among the referees. We agree with this and will include them in the manuscript and the appendix. This includes an extra panel for figure 5 with the regions of Greenland. But, I do not see an extra panel added to figure 5. As a note, I would actually suggest to combine Fig 5 and 6 as two panels in the same figure for clarity.

And, as a last thing, please carefully go through the manuscript to correct any inconsistencies and typos. Just as an example, these are some errors spotted just in the figure 6 caption
- are are -> are
- into 2 or 3 -> into two or three
- The sentence 'The background coloring is related to elevation with green as the lowest.' does not make much sense here because it is not visible.
- The southeast and east region -> The southeast and east regions
- Form -> from

I am sure that these actions will make the job much easier for the new reviewer, and you will increase your chances of a positive review.

All the best,
Louise

---

## Author Response (AR2)

Answer to the referee comment from Anonymous Referee #1

We would first like to thank the Referee for their constructive criticism of our manuscript "Sensitivity of the Greenland mass and energy balance to uncertainties in key model parameters". This is an updated answer to the referee #1, from the initial answer to the Referee # 1 published under the interactive discussion:
https://editor.copernicus.org/index.php/tc-2019-251-AC1.pdf?
_mdl=msover_md&_jrl=25&_lcm=oc108lcm109w&_acm=get_comm_file&_ms=81231&c=179958&
salt=7792715261508489784

**Summary**
The paper evaluates the sensitivity of output from a simple glacier surface mass and energy balance model to changes in its parameters. For this study the model has been extended from a previous version to include the description of turbulent latent heat fluxes, deemed important for cold climates. The evaluation is performed over the Greenland ice sheet for two contrasting climate states (present day:PD and last glacial maximum: LGM) and several regions with distinct surface mass balance regimes. The work provides detailed information about the importance of key model parameters for the surface mass and energy balance and confirms the importance of latent heat fluxes for the LGM climate.

**General comments**
The conclusions of the paper are fairly specific for this particular model. I was wondering if the manuscript wouldn't find a more appropriate audience if it was instead submitted e.g. as a model evaluation paper in GMD. This is an editorial decision, but I think it is worth considering.

We did consider GMD, but refrain from it for two reasons. Firstly, it is not directly model development, but sensitivity testing. Secondly, we wanted to raise the awareness of sensitivity and uncertainty in glaciological models.

The paper is well organised and the text is largely clear in its presentation in sections 1-2, which requires some improvement to be more precise (see detailed comments below).

My main problem with the results section (Sec. 3) is related to the challenge to present results for 9 different parameters for two different climate states, 6-12 different regions and two different analysis techniques in a concise and interesting way. I believe this section in its present form lacks focus and direction and much improvement can be made in presenting these results. More precision of the descriptions is also needed here.

I suggest the authors should look for possible generalisations across these four axes of analysis and of clear story lines that are followed throughout the discussion of results. To give an example: the manuscript discusses the notion that the Greenland margins at the LGM exhibit similar features to the interior at the PD. Maybe this can be used to generalise the results further and reduce the amount of individual cases that need to be discussed.
Overall, one possible approach could be to define a few main conclusions of the study first and then find evidence for those in the different results. If possible, consider moving less important results to an appendix or supplement.
What struck me as particularly difficult to digest are parts of the text where the description of the results happens without accompanying figures (e.g. p17.l21, p19.l17). I strongly suggest to provide additional figures to make the discussion of results more tangible. In all cases where results are not shown in figures or tables, add 'not shown' in the text, otherwise, make a reference to the figure.

We try to be more precise in the description and provide a clear distinction between the two climate states, the regions and the parameters throughout the manuscript. In particular, we will follow the referee's suggestion to use LGM and PD headings in the results and discussion section.
We added a supplement with all the different figures, and moved a clear selection to the appendix. The need for additional figures is a shared concern among the referees. This includes an extra panel for figure 5 with the regions of Greenland. A figure similar to figure 6 for the LGM period and the sensitivity of the turbulent latent heat flux as an additional figure. We added selected additional figures to the appendix. As there are two climate states, nine parameters and 11-13 regions, we created a supplement as an achieve, making it easy to find and search individual figures. Those are available at zenodoo 10.5281/zenodo.4310369. This was added to the code and data availability statement in the manuscript " The BESSI model code is available on git-hub (https://github.com/TobiasZo/BESSI/tree/TobiasZo---GSA-model-version).
Additionally, the github branch also contains the analysis and plotting scripts.
To increase the readability we included "shown/not shown" statements and references to the supplement or appendix, respectively.

Regions. It is confusing to me that the paper apparently operates with three different sets of regions (those defined in Fig2, Fig 5 and those not shown for the LGM). If at all possible, I'd suggest that one set of regions (or at least one clear definition of regions that may then results in differences between PD and LGM) be used throughout the manuscript? In the current framework, the regions defined for the LGM should be shown (they should be different from the PD if they are based on elevation) as results are given for those. It seems that the PD analysis is limited to region 5 only, so it is a surprising choice to show all PD regions in detail, but none of the LGM regions.

We provide a clear definition for the 11 respectively 13 regions. They are defined geographically by the present day ice divides and elevation. Though due to the different ice sheet topography during the LGM, they are similar but not identical. The regions in Fig 2 are chosen to be only based on elevation to keep the focus on the GSA method and its uncertainty rather than a regional definition. We want to introduce the concept rather than providing in depth analysis, which is done based on figure 3 and 4 for the GSA. Regions solely based on elevation do not take into account the large climatic differences between parts of Greenland. Therefore the GSA analysis is mainly based on a distributed approach shown in Figure 3 and 4.

As pointed out in the text, the relative importance of different parameters in the global sensitivity analysis are dependent on the sampled parameter ranges. I miss a clear motivation and argument for the plausibility of the assumed ranges beyond reference to Born et al. (2019). This seems like an important aspect of the paper, so it should get some attention in the text. This includes the question if the parameter ranges supposedly derived for PD climate hold for the LGM and what could be done to mitigate this effect, if any.

We added literature references to Table 1 regarding the parameter range. Additionally, we extend the discussion of the dependency on the range in the respective session of the manuscript. The only clear change from parameter range used in Born et al. (2019) is in the snow albedo, this is due the implementation of additional albedo schemes where a wider range is plausible than in the two cases one used in that study.

**Specific comments**
Up front a few points that are repeated throughout the manuscript. Some examples are given below.
- Reconsider the use of 'it' and 'this' in cases where the subject of the sentence is not clear.

- Distinguish between physical quantities and mechanisms on one hand and the parameters that influence those on the other hand.
- Be clear about what is shown in a given figure (PD or LGM) and what results you are discussing in a given paragraph (PD vs LGM). Possibly use section headers to make that distinction.
- Be explicit what you are comparing in a given part of the text: PD vs. LGM, one region against another or one parameter against another.
- In the context of this paper, I would always refer to the contrast between PD and LGM as 'difference' rather than 'change', since there is no time dependence here.
Title: Add 'surface' before 'mass and energy balance'

We incorporate these comments as much as possible. We changed the title to "Sensitivity of a Greenland surface mass and energy balance to uncertainties in key model parameters".

Abstract: Could add more information about the model and experimental setup.

We did not add substantial information about the experimental setup to the abstract.

p1.l2 remove 'climate' after 'present day'. A climate is not a period.
p1.l16 If this is supposed to be a reference for the ITM method, more appropriate references may be Bintanja et al., 2002 or Van den Berg et al., 2008
p2.l2 Add 'computationally' before 'too expensive'.
p2.l9 Maybe 'can be used', to make clear they are not used in parallel.

We include all suggested changes.

p3.l9 Not obvious what a 'mass following' grid is. Clarify, add a reference.
p3.l9 Add 'in the snowpack' after '15 layers' if that is the correct description.
p3.l9 Maybe 'The mass of each layer is 100 - 500 kgm-2'. Clarify if the mass is decreasing/increasing with depth or where the range originates from.
p4.l12 We don't know yet how large the boxes are! Also, you use the term 'layers' before. Is large the right term for a layer defined by its mass?

The vertical grid is defined by mass and not height. We will clarify this in the methods section and make a clearer reference to Born et al. (2019). The mass of each layer is 100-500 kg/m2. Each cell is initially filled up to 300 kg/m², but due to melt and refreezing the mass may in- or decrease. Cells above 500 kg/m² or below 100 kg/m² are split or merged respectively. We are referring to a large box, meaning being thick. We add max and min estimates to the particular statement (0.2-1.4m).

p3.l12 Clarify what happens to the other variables if they are not downscaled to the model topography. Are they just interpolated?
p3.l17 Which two? The last two? Clarify

All variables are interpolated bi-linearly to the horizontal model grid. Only the atmospheric temperature is corrected for the actual model topography that is generally different from that of the input data. Only precipitation and turbulent latent heat flux are associated with a direct mass change on the RHS of the equation.

p3.l24 Insert 'then' after 'The actual melt is'. flux
p4.l1 Consider adding numbers for the 4 different parametrizations below. 1. Constant, 2. Oerlemans and Krapp …

p4.l12 The end of the sentence 'would likely already be wet if the real surface was resolved' needs further explanation to be comprehensible.
p4.l14 Consider using same notation as for the other forms (number) author: ...
p4.equ(4) Add 5d for Ts = 273K?
p4,l21 Clarify what the first 'this' refers to in 'this was adapted for this model'. Clarify what 'fixed and temperature dependent' means. How can it be both at the same time?
p4.l25 Clarify what 'it' refers to in 'Keeping it constant'

We included the changes and clarify where necessary. The terms 'fixed' and 'temperature dependent' are misleading and are rephrased. See the changes markup file for the details

p5.l7 Could explain physically what this approach is trying to mimic. Supposedly that the first snow that falls on a wet surface will get wet immediately.

This albedo increase always depends on the amount of snowfall, a few cm of snowfall will not lead to full fresh snow albedo as solar radiation penetrates and the darker old snow is still visible at the surface. Therefore we have an incremental increase with the amount of snowfall. Albedo reseeding is present at every snowfall, but if there is still liquid water present in the layer the albedo will be decreased depending on the liquid water content. We are not resolving standing water at the surface, but only have the liquid water content of entire grid cells.

p5.l21 'ice-sheet' –> 'ice sheet'
p6.l16 Typo 'fesetup' –> setup
p6.l24 Insert 'in the temperature snowpack' after 'grid cell'. Consider consistent use of terms 'layer' vs. 'grid cell' vs. 'grid box'.
p7.l12 'The global sensitivity analysis is a variance-based method'
p7.l13 'In contrast to other methods'
p7.l13 'all parameters are varied at the same time'
p7.l15 remove 'using' after 'hypercube'.
p8.l19 'As the ice sheet has different shapes' –> 'As the ice sheet geometry differs between the two climate states' or similar.

All changes accepted.

p6.l23 'heat supplied by precipitation depends on the temperature'. Which temperature?

"atmospheric" was added for clarification.

p7.fig1 Why is the colour over the ocean different in a and c? Why is there a contour line in the ocean? Are elevations at LGM relative to LGM sea level (as they should) or relative to PD sea-level? Suggest to plot all topographies with ocean surface at sea-level = 0.

During the present day we have the sea floor also resolved, we will change this by setting the ocean surface to 0°C everywhere. References are relative to the sea level at that time.

p8.l18 Specify temperatures in degree C instead.

We changed the temperature to °C everywhere during the climate discussion.

p8.l19 Why not use the larger ice sheet area of the LGM for both to avoid biases e.g. from the large ocean area in the south-east?

Biases arise in either case as the actual ice sheet extent is not similar. The statement is just taken as a rough comparison between the climate states, therefore we do not provide uncertainties and just describe how it was calculated.

p8.l25 How is the model run back and forth? Is it reversible problem?  in time?

We will clarify this in the revised manuscript. "BESSI was run for 500 years with the same forcing data looping the forcing data back and forth (1979-2017-1979-2017…)." This effectively avoids unrealistic jumps in the boundary conditions that would otherwise arise from the (temperature) trend in the observational period. The model itself always runs forward in time.

p8.l31 Explain why the ensemble is split in two.

We are going to include the following description in the revised manuscript: The initial ensemble was generated using a Sobol sequence which consisted of 2000x9 members for PD and 1000x9 for the LGM. This sequence spans a 9-dimensional unit hypercube. For computing both sensitivity indices the estimator from \citet{Sobol2007} was used. It splits the initial sequence into two subsets A B each consisting of one half of the initial sequence (1000/500x9). Then an additional set of matrices $B\_A^i$, which are based on the matrix B where the values for parameter for parameter $X\_i$ are replaced with those from subset A, are created.

p9.l1 Suggest to move 'A detailed description of the algorithm can be found in (Sobol et al., 2007) and (Saltelli et al., 2010).' after 'used to estimate the model sensitivity.' to where the general description of the method ends. Also, specify k in your example.
p9.l15 Insert 'STi' after 'total sensitivity index'.
p9.l16 Clarify temporal or spatial average in 'surface mass balance and the average surface temperature'.
p9.l17 Specify which variables are averaged and which are summed.
p9.18 'reseeding' –> 'residing'

We adopt all suggested changes and clarifications.

p10.fig2 Do you explain somewhere how the sensitivity is normalised? The plot of Greenland looks distorted. Can this be plotted in equal aspect ratio. Additional (white) contour lines could make the region separation clearer. Caption: Add panel for 'entire ice sheet' to the description. Consider adding panel indicators a-f for the boxes and g for the GrIS plot and use them in the text.

Equation 17 and 18 explain this. The plot is slightly distorted as the focus on the aspects was on the content of the graphics.. We will not add white contour lines, the focus of this plot is the sensitivity analysis and its uncertainty by elevation, we want to avoid focus on regional differences. The map serves as an indicator for the elevation bands. We added the indicators a-f.

P10.l4 Maybe 'low' –> 'limited'
p10.l4 Remove 'changes' after 'SMB'
p10.l5 See general comment on relative sensitivity of parameters dependent on choice

of parameter range. Should be clarified here.
p10.l6 Check consistency Eatm (here) - Eair (Figure 2)
p10.l7 Check consistency DSH (here) - Dsf (Figure 2)
p10.l9 Add 'albedo' after 'the fresh snow'

We apply the changes and check for consistency.

p10.l14 is 'above 2000 m' > 2000 or the region 2000-3000? Clarify

This was changed to "regions above 2000 m".

p11.fig3 Title: 'Sensitivity of the SMB at PD' Are values >1 actually defined for this
analysis? If not, why is the yellow arrow on the colour bar? What does a sensitivity
index above 1 mean? The colour scale with the darkest colour for the least important
results is not convincing me. Contour lines are not visible on most plots. Suggest a
different (lighter) colour and omitting the numbers. Caption: Mention figure is for PD.
Mention colour choice for ice free land. 'mass balance' –> 'surface mass balance'.
Reformulate 'are not to be taken too seriously'. If we don't take the absolute values
seriously, what is left in this figure? If relative values are more important, find a way to
plot those instead.

Total sensitivity indices, which include interactions could mathematically be greater than 1 due to it
being only estimated. The sum of the total indices can be greater than 1, while the main order
indices cannot. We will change the contour lines and adjust the figure caption. Though we will
keep the color coding. The wording of seriously is unfortunate and will be changed. We wanted to
highlight that these numbers similarly to figure 2 come with uncertainties, but we wanted to avoid
showing two additional uncertainty figures.
We created an alternative figure coloring keeping the old colorbar, but changed the contour lines
to white. Actually, due to an overlap with the white background of the ocean, we colored that one in
light blue instead. Red contours would have worked best, but we assume that the greenish-red
combination does not work for colorblind people. This is added as extra figures, we would suggest
the editor may take a final decision.

p12.l2 Reformulate 'the structure is much more complex'. What does that mean? There
is more information in 2D compared to 0D?

We changed this to "... but there is also a spatial dependency which is not purely elevational".

p12.l3 remove 'glaciers' after 'west coast'.

We are actually referring to small ice caps and glacier cells on the west coast of Greenland south
of Disko Bay. We removed the statement as it does not serve any discussion purpose, though the
feature is clearly visible in the GSA map.

p12.l7,8 2x 'becomes' –> 'is'. Otherwise this implies important a process.
p12.l30 Reformulate 'negative ensemble member'.
p12.l33 'interior of Greenland

We adopt the changes.

p12.l33 Clarify 'the atmosphere is more in balance with the snow surface'. What does that mean?

It refers to "the temperature is much closer to the snow surface temperature".

p12.l34 Consider adding a section header 'LGM analysis' or similar to make a clearer separation between the two climate states in the text.
p14.l8 Would an additional figure similar to Fig2 for the LGM help?
p14.l2 'The ice sheet integrated SMB'. Do we see this somewhere? Consider adding a figure or table and refer to it here.

See general comments. We include a similar plot as fig 2. in the Appendix as A1.

p13.fig4 Title: 'Sensitivity of the SMB at LGM' See comments for fig3 for general layout.
Caption: Start with description what is shown, not with discussion of the figure.

We use a similar caption description as for Figure 3.

p14.l4 'do not impact the SMB'. Not at all? Clarify

We changed the wording to "marginally".

p14.l4 add 'the' in 'either of the two climate states'. Refer to fig 3 and 4 then.
p14.l4-7 First start discussion of LGM, then continue comparison LGM-PD.
p14.l8 'increased'. Relative to PD? Clarify.
p14.l9 Add 'surface' before 'mass balance'.
p14.l11 'intra-annual variability' –> 'seasonal variability'
p14.l15 remove 'temperature' after 'surface and the air', to avoid duplication.
p14.l16 'the fewest precipitation amounts'. Is this shown somewhere? If not, add 'not shown' in the text.
p14.l18 Add 'at LGM' after 'The reduced model sensitivity'.
p14.l21 'The sensitivity of ...' to what? maybe 'model parameters'?

We include the suggested changes and clarify where necessary.

p14.l12-14 'the ... impact of ... impacts the latent heat flux more than the actual exchange coefficient'?? Not clear. Reformulate. Clarify.

We changed this to "...because the surface temperature via the Clausis-Clapeyron relation has an exponential impact on the latent heat flux resulting in a greater impact than the actual exchange coefficient."

p14.l22 'without additional figures'. Not sure this is a good idea, see general point.
Maybe add a table with results instead?

We cannot show 2D data in a table in a reasonable fashion. Therefore, we include a figure in the appendix showing at least the sensitivity of the turbulent latent heat flux (A3).

p14.l23 What is 'final firn albedo'? Reformulate
p14.l24-25 Not sure I understand this conclusion. Reformulate?
p14.l26 What does 'it' refer to in 'it is most sensitive'? Clarify
p14.l28 'as are the snow albedo related ones in the north' –> 'as are the ones related to snow albedo in the north'
p14.l30 Start sentence with 'Globally' and then give specific details in the end.
p14.l31 'the framework' , maybe 'this framework', 'our framework'

We include the suggested changes and clarify where necessary.

p15.fig5 Add panel with regions for LGM, which are different and actually used in the analysis (unlike only region 5 for PD). Why invent new regions after what is already introduced in figure 2? Could you not do the analysis for region 1000-2000 instead of region 5? Or find a common subset to use in fig 2? Caption: Start caption with what is displayed in the figure. Results are for the text.

We are including a second panel for the LGM. The regions are defined based on the same present day ice divide and the elevations respectively. Our approach is to start from simple elevation based regions and then go more into detail of more complex patterns. We cannot use 1000-2000 m as one region as the SE is too different from others for example. Region five is 1000 -2000 m at the west side. The regions used here are chosen based on similarity, but we don't want to overcomplicate figure 2 with introducing this already. It makes it easier for the reader to start with known regions. We initially had 28 regions with four elevation bands and 7 geographical areas, we joined them based on similarity in sensitivity of the surface mass balance.

p15.l1 'more' or 'less' than what? Maybe 'closely linked to the SMB'?
p15.l1 'and shows similar sensitivities as have been reported for the SMB'
p15.l4 Add 'surface' before 'mass balance'.
p15.l9 'and 13 for the LGM (2 more around Elsmere Island)'. Need to show these in a figure.

We include the suggested changes and clarify where necessary.

p15.l2 'much lower.' compared to what? Why is that expected? Explain.

Just as expected, the impact of the latent heat flux switch on the snowmelt is much lower than on the SMB.

p15.l6 'Parameters which result in either surface heating or cooling'. More precision needed. It is not the parameters that result in heating or cooling, but the physical process that is parameterised.

We change this to "physical processes which result... like the turbulent fluxes and the associated parameters".

p15.l7 Add 'Conversely' before 'Albedo', as these are examples where GSA will work.

p15.l8 Add 'so that GSA gives clear results' or similar after 'mass balance'.
p16.l4-6 According to you, parameter Eatm can be well analysed with GSA. Why does
it need more detail here?

We want to clarify that GSA always works, but there is additional information extractable with other analysis.

p15.l19 'based on elevation and similarity'. Similarity of what? Explain.

See the comments to figure 5.

p16.fig6 Add yticks in column 2 and 3. Suggest to plot all results as discrete boxes. The
mix between continuous quantiles and discrete outliers looks strange to me. Caption:
Add that this is for PD.

Discrete boxes do not work for this amount of boxes, we will try to plot the outliers at their actual location rather than the center of the box.

p16.l1 What does 'It' refer to in 'It shows'?

We added Fig 6 (a-i).

p16.l6 'accelerating manner' could suggest that the parameter should be sampled non-linearly.

Yes, for research on a particular region or point in Greenland sampling the parameter space non-linearly makes sense, but in this study we investigate the entire ice sheet. Therefore, it is not feasible.

p16.l8 ' the width of the distribution decreases' Could you explain why this is the case?
I would think with more available energy (l7), differences in the other parameters have
a larger impact on the SMB. Similar with lower albedo (l8), differences in the other
parameters are more effective in making a difference.

This is exactly the case. If there is already high energy input (through low albedo or high atmospheric emissivity), and the surface is close to or at the melting point the SMB is much more sensitive to changes in other components of the SEB.

p17.l4 'has the highest median mass balance'. Explain why.

This is explained in line 15 ff.

p17.l6 'The strong impact' on what?
p17.l7 'the other fluxes'. What kind of fluxes? Energy?
p17.l21 'During the LGM the western region between 1000 - 2000 m'. Should be
shown.
p17.23 'three distinct changes' –> 'three distinct differences relative to PD'
p17.l23 'χQL results in a decrease of SMB'. Distinguish physical process and parame-
ters.

We include the suggested changes and clarify where necessary.

p17.l26 'slower snow albedo decay'. This is discussed as a general result, but is not available in all albedo models, is it?

Though the meaning of the statement is not wrong, we added "for all albedo subroutines which incorporate decay" at the end of the statement.

p17.l30- Make clear that the discussion is back to LGM results.

See general remarks.

p17.l28 'sublimation ... results in a mass loss rather than a mass gain as in PD.' You should probably distinguish the opposite sign using the term 'deposition' or 'desublimation'.

No, we are not talking about deposition or resublimation. Due to the sublimation the surface cools which results in less melting and a more positive SMB.
We removed the parenthesis for clarification. "During the LGM sublimation prevails over the entire year, but in the absence of melt it results in a mass loss rather than a mass gain as in PD via cooling and associated reduced melt."

p18.fig7 Panels are difficult to compare due to different vertical scale. Add xticks in row 2 and 3 Caption: Add that this is for LGM. Highest elevation at bottom is counterintuitive, consider changing order. Add figure with regions and link from here.

This is not for the LGM, but we add PD for clarification and adjust the caption accordingly.

p19.l2 'The smallest spread of the ensemble is found in the high altitude-regions 9, 10, 11'. Difficult to judge with different vertical scales in figure. Also, this is a new point. First finish discussion of DSH?

We changed the statement to "smallest relative spread"

p19.l3 'a result of higher air-temperatures than snow surface temperatures'. Needs more explanation to be clear.

Added: "the warming effect that D_SH has on the surface due to on average higher air-temp."

P19.l7 What is 'moisture differences between surface and atmosphere'?

We change this to the water vapor pressure of the surface and the atmosphere.

p19.l8 '7-11' was 9-11 in the explanation above at l2. Clarify.

The first refers to narrower, while the other mentions smallest.

p19.l14 'it acts as a buffer of the SMB'. Not clear.

At strong turbulent sensible heat exchange the surface temperature will be buffered by the air-temperature heat reservoir.

p19.l18 Add 'Ql' after latent heat flux.
p19.l17- Hard to follow this part without any guidance. Include a figure?
p19.l28 'the SMB decreases' with what?
p19.l29 'behavior to the GSA' –> 'behavior as shown in the GSA'
p19.l35 'Similar to the GSA' –> 'Similar to the results from the GSA'
p20.l1 'above 2500 instead of above 3000 m'. Is this comparing to PD? If so, mention it.

We adopted the changes. A figure will be included for the turbulent latent heat flux (A3).

p20.l6 'in region 5'. And in in all the other regions? No discussion of those?

We focus on region 5 as it shows interesting features. All other regional data will be made available.

p20.l8 'the air-temperature buffers the snow temperature'. Not clear. Clarify.

At strong turbulent sensible heat exchange the surface temperature will be buffered by the air-temperature heat reservoir.

p20.l14 'The model sensitivity in this study is evaluated' –> 'The model sensitivity is evaluated in this study'
p20.l15 'big' –> 'large'

p20.l17-18 I would say it is the other way around: lower atmospheric temperatures ...
'leading to fewer areas where melt and runoff occurs' and consequently 'reduce the impact of QLWin and εatm

At lower atmospheric temperatures the net SEB gets more negative (i.e less heat input from the atmosphere) due to a reduction of QSH and QLWin. Less energy will result in colder snow temperatures, reduced melt and melt area extent.

p20.l20 'This is due to the absence of melt'. I read this as complete absence of melt. Is that correct? Otherwise 'negligible amount of melt'?

We add in large parts of the ice sheet, as this seemed ambiguous despite mentioning it in the sentence before.

p20.l26 Is it correct that the long-wave radiation is twice as large as the incoming solar radiation? This is for heavy cloud cover?

These are annual averages for coastal areas. So for areas with higher cloud cover, but not for individual days of heavy cloud cover.

p20.l25 'bigger' –> 'larger'
p20.l25 'for the surface during PD' –> 'for the surface energy balance at PD'
p21.l1 Add 'surface' before 'mass balance'.
p21.l12 'the temporal change of the sensitivities'. What is that? The contrast between LGM and PD?
p21.l15- Try to formulate this positively to make it clearer.
p21.l18 Replace 'increased' by 'larger'
p21.l19 'during the LGM' –> 'at the LGM'. Similar for PD and in other places.
p21.l19 replace 'increase' by 'influence'.
p22.l4 Move 'climate' to after 'glacial maximum' in the next line.
p22.l5 'study the change of the model response under different boundary conditions' –> 'study the differences of the model response under LGM and PD boundary conditions'.
p22.l6 add 'for the LGM climate' after 'is a necessity'
p22.l9 'creates a SMB model uncertainty' –> 'govern the SMB model uncertainty'
p22.l9-10 'With the change in circulation during the last glacial a changing energy input from the atmosphere to the surface will result in a SMB response' –> 'With the different circulation during the last glacial maximum a changing energy input from the atmosphere to the surface will result in a SMB difference.'

We considered all suggestions mentioned by the referee.

p22.l11-14 I don't think this has been shown in the manuscript. This could be a discussion item, but not a conclusion of the manuscript if it is not even mentioned before.
p22.l16-18 This reads like a discussion item, not like a conclusion.

We moved this part to the discussion.

---

## Author Response (AR3)

Answer to the Referee:

This manuscript investigates the parameter sensitivity for a glacier surface mass and energy balance model. I'm less familiar with this model, which calculates energy balance using just temperature, precipitation, and incoming shortwave radiation as inputs, and parameterized other components, but the model is well-described. Authors perform experiments for parameter sensitivity over Greenland for two time periods: the present day and Last Glacial Maximum (LGM).

The introduction sets up the paper well. I think that the experiments in this paper, showing which parameters mass balance is most sensitive to, are important and will be useful for others simulating mass balance for glaciers and ice sheets. The presentation of results isn't always intuitive, and at times the manuscript does not clearly and accurately communicate the key messages.

We thank the referee for the thoughtful comments.

Specific comments

p1 L4: Is 'invariant' the correct word? You show that the sensitivity varies by region (Fig 3).

We fully agree that it is the magnitude of sensitivity change that are similar in space and time. We changed it to: "The sensitivity towards individual model parameters and parameterization are as variable in space as in time". Which also includes the case of the 3 "non-sensitive" parameters which are just not variable at all.

P1 L4: Should you clarify that emissivity has, by far, the highest sensitivity (at least in present day)?

Changed to include the emissivity and the long-wave radiation in this sentence.

P1 abstract: Is it useful to note differences between present and LGM? The comparisons seem like a big part of the paper.

We changed the sentence about the turbulent latent heat flux, so it puts higher emphasis on the difference between the LGM and PD.

P1 L22: The use of the word 'unfortunate' here feels like an opinion. This occurs throughout the text, with words like 'rather' and 'clearly'. I think these words should be removed. We changed the wording of unfortunate and removed multiple usages of the other words.

P9 Table 1: Does parameter #5 have a citation?

No it does not have one directly. As it was not used in these relative terms, though Rolstad et al. and citations within discuss the differences of the exchange coefficient for sensible and latent heat flux.

P9 Table 1: Parameter #8, I think a different abbreviation is used later in Fig 2 (am). Keep these consistent. We added the full names to the table.

P10 L16: Lots of acronyms here. Better to say 'Present' instead of PD? We keep the acronyms through out the text, but repeat "present day (PD)" at the beginning of every section now.

P10 L21: remove 'a slight influence of' We followed the referee's suggestion. Fig 2: Great, this seems like a key figure. It's not clear what the difference between main and total effect are. Did I miss this in the methods? Also, as a key figure, why is the equivalent for LGM not in the main text? It seems like it would be more useful to have that figure in the text, and current Figures 1 and 5 either smaller or in the supplement.

Figure 3 and 4 do show the sensitivity indices on a distributed scale, therefore we show those two for both climate states (PD, LGM) and consider it as the key figures. The additional gain of figure 2 are the uncertainties of the GSA method, but we consider it not necessary to show these for both states at the main part of the manuscript.

The sensitivity indices are described in the methods, we added a reference and a brief description to the figure caption to make it easier to understand.

P15 L6-7: Delete 'Due to their large size,' and 'but we include the main findings here'. We followed the referee's suggestion.

P15 L30: I'm confused here. Maybe just 'Conversely' needs to be removed? We want to show the contrast between parameters that are directly correlated with increase and decrease, and those that may give either depending on the environmental conditions. We changed the wording to "on the other hand".

P 16 L 5: Why pick region 5? Later you mention the ELA is within the region, maybe include that here.

We added a statement about our decision concerning the region and the ELA.

Figure 6: Text says that a) and b) show sensitivity, but not c) (P16 L 10-14). But they all look pretty similar to me. Maybe labeling SMB on all axes would help. There is a trend in a and b and not in c. The plot without shared y-axes is too crowded and complicates the readability.

Figure 6: Why are the lightest color vertical bars, while the 3 darker colors are continuous? These are the outliers of the 90% range and are plotted at the center of each bin.

P18 L 22-25: Really hard to follow the list, adding numbers in front of each point could help (e.g. 1) The impact of ...) We followed the referee's suggestion.

Figure 7: I'm a bit confused here, you're looking at the sensitivity of parameters on surface mass balance only with QL on, still for region 5 at the LGM? I would expect it to be the equivalent to Fig 6 but for LGM, but what is going on with QL?

If we only consider the sub-ensemble where QL\_on is present than there are no members which have QL\_off, therefore the plot for X\_QL has no "left" box.

Figure 8: This is figure 8, typo in caption? Thanks for noting the referencing error figure 7 was meant.

Figure 8: Why did you choose to show parameter sensitivity by region only for Dsh? Discussion: I found this section a bit hard to read. Some sentences might need commas to make the point clear.

We focus the discussion on this parameter as it may lead to a higher or a lower SMB depending on the atmospheric conditions. We added an additional line, referencing the supplement for further figures for the other parameters.

The whole section was reworked to make it easier to follow.

P23 L12: delete 'while' and rewrite? There is still sensitivity to emissivity during the LGM. We changed this to "During the LGM the SMB shows additional increased sensitivity to the fresh snow albedo, the choice of albedo parameterization, and the turbulent latent heat flux."

**P23 L15: what does 'though desirable' mean?**

Uncertainties in SMB related to this parameter are small, so we conclude that it is not necessary to include it, but better to do so.

P24 L8: Change 'neither' to 'None', as you have multiple variables? We followed the referee's suggestion.

P24 L16: Do you show you improve the model with these new additions? No, we do not show it in this manuscript, but there was a previous bias in dry regions due to a lack of sublimation (Born2019, Imhof 2016) which is removed now.

P24 L 23-25: Is there a previous description of the different circulation during the LGM? Having it just brought up here is missing an explanation if one doesn't know about it. We added a connection to the results and discussion section about the circulation changes due to the Laurentide ice sheet.

Appendix: Some plots mentioned in the text are missing, or is there also a supplement? If so, why are some in Appendix and some in supplement?

There is supplement which is referenced together with the model code. "The BESSI model code is available on git-hub (https://github.com/TobiasZo/BESSI/tree/TobiasZo---GSA-model-version). Additionally, the github branch also contains the analysis and plotting scripts. It is also available together with the supplement at 10.5281/zen-odo.4310369."

There are a total of 176 figures in the supplement, we systematically named them so searching for particular figures is easier than having on supplement document. Only figures that are discussed in more detail are in the main manuscript or the appendix.

Born, Andreas, Michael A. Imhof, and Thomas F. Stocker. "An efficient surface energy–mass balance model for snow and ice." *The Cryosphere* 13.5 (2019): 1529-1546.

Imhof, Michael. *An Energy and Mass Balance Firn Model coupled to the Ice Sheets of the Northern Hemisphere*. Diss. Universität Bern, 2016.

Rolstad, C. and Oerlemans, J.: The residual method for determination of the turbulent exchange coefficient applied to automatic weather station data from Iceland, Switzerland and West Greenland, Annals of Glaciology, 42, 367–372, 2005